# Multi-modal dissection of cell-type specific TDP-43 pathology in the motor cortex

Wolfgang P. Ruf[1,2,9], Julia K. Kühlwein[1,9], Laura Meier[1], Sarah J. Brockmann[1], Jaehyun LeeBae ®[2], Ghazaleh Sadri-Vakili[3], Deniz Yilmazer-Hanke ®[4], Susanne Petri[5], Dietmar R. Thal ®[6,7,8], Veselin Grozdanov ®[1,9] ✉ & Karin M. Danzer ®[1,2,9] ✉

Cytoplasmic TDP-43 pathology is a pathological sign of ALS/ALS-FTD and a converging disease event across different genotypes, phenotypes and CNS areas. To understand this process and target it therapeutically, we need to define which cell types are affected and which cell-type specific effects make them particularly vulnerable. We coupled flow-cytometry nuclear sorting and sequencing with single-nucleus multi-omic ATAC-seq and RNA-seq and spatial transcriptomics to define the transcriptional cell type of affected neurons in the post-mortem ALS/ALS-FTD motor cortex (30 ALS, 20 ALS-FTD & 32 control samples). Here, we show that mainly excitatory cortical neurons are affected by TDP-43 pathology and define the cell types that are affected the most: intratelencephalic L2-L3-LINC00507-FREM3, L3-L5-RORB-LNX2, L3-L5-RORB-ADGRL4 & L6-THEMIS-LINC00343 neurons and extratelencephalic L5-FEZF2-NTNG1 neurons. Transcriptional aberrations by TDP-43 pathology, like cryptic exon inclusion, are cell-type specific and affect distinct gene sets in each cell type, highlighting the need to address TDP-43 pathology in a cell-type specific manner.

Amyotrophic lateral sclerosis (ALS) is a devastating neurodegenerative disease that affects the brain and spinal cord and rapidly progresses to severe disability and fatal end[1,2]. While ALS is primarily characterized by motor symptoms, up to 50% of ALS patients display an overlap with the cognitive disability profile of frontotemporal dementia (FTD), another devastating neurodegenerative disease affecting the frontal and temporal lobes of the brain[1–3]. Together, ALS and FTD represent two ends of a spectrum of neurodegenerative disorders ("ALS-FTD") that share clinical, genetic and pathological features[4,5]. ALS-FTD is associated with the degeneration of specific populations of cortical neurons in different regions of the frontotemporal cortex, including

the giant pyramidal Betz cells and the von Economo neurons[6,7]. The pathologically defining feature of degenerating cells in ALS-FTD is the presence of proteinaceous cytoplasmic inclusions rich in ubiquitinated aggregates of the protein TDP-43 (TARDBP gene) that can be observed in > 95% of the ALS patients and > 50% of the FTD patients[1,7–9].

TDP-43 aggregation is believed to not only represent a pathological correlate of degeneration, but also to play an active role in the degenerative process. Indeed, translocation of TDP-43 to the cytoplasm and aggregation results in different cellular aberrations, including de-repression of transposable elements[10], expression of endogenous retroviral proteins[11], erroneous splicing and inclusion of

[1]Department of Neurology, University Clinic, University of Ulm, Ulm, Germany. [2]German Center for Neurodegenerative Diseases (DZNE), Ulm, Germany. [3]Sean M. Healey & AMG Center for ALS at Mass General, Massachusetts General Hospital, Boston, MA, USA. [4]Clinical Neuroanatomy, Department of Neurology, University Clinic, University of Ulm, Ulm, Germany. [5]Department of Neurology, Hannover Medical School, 1, Carl-Neuberg-Strasse, Hannover, Germany. [6]Laboratory for Neuropathology, Department of Imaging and Pathology and Leuven Brain Institute (LBI), KU-Leuven, Herestraat 49, Leuven, Belgium. [7]Department of Pathology, University Hospitals Leuven, Leuven, Belgium. [8]Laboratory of Neuropathology, Institute of Pathology, Ulm University, Ulm, Germany. [9]These authors contributed equally: Wolfgang P. Ruf, Julia K. Kühlwein, Veselin Grozdanov, Karin M. Danzer. ✉e-mail: veselin.grozdanov@uni-ulm.de; karin.danzer@dzne.de

cryptic exons (*CE*) [12–14], altered poly-adenylation site usage[15], changes in gene expression[10] and cellular stress. These molecular aberrations are well-described, and some of them are 'upstream' of other ALS-FTD genes, like *STMN2*[16]. In addition, it is also likely that pathology spreads from a neuron to its adjacent and to its interconnected cells, affecting progressively more cells and areas. However, only a subset of the brain cells is affected by TDP-43 aggregation and degeneration in ALS-FTD, a phenomenon termed 'selective vulnerability'. The molecular basis and the genetic/epigenetic predisposition factors for the selective vulnerability are not well understood.

Histopathological studies have demonstrated that not only the 'canonical' vulnerable neurons (*i.e., Betz cells, the L5b extra-telencephalic neurons*), but also neurons from different cortical layers are affected by TDP-43 pathology[7]. However, the molecular and transcriptional identity of these neurons have not been defined, despite three recent attempts: Liu et al. could demonstrate enrichment of upper layer (*L1-L3*) neuronal markers in bulk RNA-seq of TDP-43 depleted nuclei from the middle frontal cortex[10], while Gittings et al. and Wang et al. could successfully impute TDP-43 pathology of single-nucleus RNA-sequencing, but remained limited to the largest affected group of L2-3 ITC neurons, both in the dorso-lateral prefrontal cortex (*DLPF*)[17,18]. Recent advances in single-nucleus RNA-sequencing have enabled the highly resolved description of different cortical cell types[19]. Still, the transcriptional cell types affected by TDP-43 pathology and its effects in them have not been defined yet. The targeting of specific molecular processes upstream or downstream of TDP-43 pathology requires the precise resolution of the cell-type identity and of the transcriptional events.

Here, we characterize the cell-type specificity of TDP-43 pathology in the primary ALS/ALS-FTD motor cortex by employing a multi-modal approach that enables highly-resolved identification of cell types and direct identification of single nuclei with TDP-43 pathology. We found that TDP-43 pathology affects predominantly specific cell types and results in distinct transcriptional alterations. These effects were cell-type specific and distinct from the general transcriptional changes in the ALS/ALS-FTD motor cortex, which were partly associated with differential chromatin accessibility.

## Results

### Multi-modal ATAC-seq and RNA-seq in the same nucleus reveals an increased resolution of cell-type identity in the motor cortex

We generated a multi-modal, multi-cohort single-nucleus dataset of the diseased and unaffected human primary motor cortex ('M1') by sampling fresh-frozen postmortem tissue from 30 patients with sporadic ALS ('ALS'), 10 patients with sporadic ALS-FTD ('ALS-FTD'), 7 familial ALS-FTD patients harboring the pathogenic *C9ORF72* Hexanucleotide repeat (HRE) ('ALS-FTD C9ORF72'), and 32 neurologically unaffected controls ('Ctrl', study design in Fig. 1a) from a total of six different cohorts. ALS and ALS-FTD were diagnosed based on the standardized criteria at the respective centers and pathologically confirmed by TDP-43 inclusions *post mortem* (*47/47 cases*, Supplementary Fig. 1). Classification into sporadic/familial disease was based on medical records and a genotyping panel testing for pathogenic variants in 43 ALS-associated genes (*clinical and demographic characteristics of the cohort in* Supplementary Data 1 and Supplementary Data 2, *list of tested ALS-associated genes in Methods*). By utilizing the 10X Genomics Multiome Assay for Transposase-Accessible Chromatin by sequencing (ATAC) + RNA-seq Kit, we acquired a total of > 245,000 nuclei with simultaneous chromatin accessibility and transcriptome profiles, resulting in >180,000 high-quality nuclei after thorough quality control ("Methods", Supplementary Fig. 2). To alleviate confounding by batch effects, we applied an experimental design where control and disease case samples were processed together in different 10X Genomics 3'GEX reaction wells and some samples repeated on different 10X reaction wells (Supplementary Data 3). We then integrated all samples into a fully-merged, integrated dataset by carefully modeling the batch effects with deep learning (*scVI for GEX and peakVI for ATAC*), aiming for a robust clustering of nuclei to cell communities independent from batch and the donor disease status. To reveal cell identity and group cells with similar chromatin accessibility profiles and transcriptome programs, we leveraged both RNA and ATAC modality by employing the weighted-nearest-neighbor (*WNN*) framework. This combines latent spaces from chromatin accessibility and the gene expression modalities into a single latent space, accommodating the need for a single representation of the dataset in a reduced dimensionality space. This approach enabled the fine resolution of the nuclei into discrete cell types that could not be identified using either of the modalities alone (Fig. 1b–d and Supplementary Fig. 3), with 58 cell clusters identified from the transcriptomic profiles alone, 64 with the chromatin accessibility profiles alone, and 75 with WNN. WNN nuclei clusters were assigned to a cell type by marker expression and then grouped hierarchically into major cell types and classes (*see next section*). All major human cortex cell types (*excitatory neurons, inhibitory neurons, astrocytes, microglia, oligodendrocytes and oligodendrocyte precursor cells*) were detected, with a strong overrepresentation of excitatory neurons, as expected for proper sampling from the M1 area. Vascular, epithelial, and peripheral immune cells were also detected, but due to their very low numbers that prevent statistically robust conclusions (*1,045 nuclei, 5 clusters*), we excluded them from further analysis. All major cell types were similarly represented in each disease group (Figs.1e, 2e), with significant variation between the samples. The major cell-type WNN clusters were well-characterized by marker features from both modalities (Fig. 1f, g), confirming the proper segmentation of the nuclei into major cell classes.

### Hierarchical annotation of cell-type identity and compositional analysis

To leverage the full power of the single-nuclei dataset, we modeled cell identity with a hierarchical cell-type annotation, which allows for the balancing between the advantages and limitations of single-nucleus resolution. We grouped the 70 cell clusters (*75 total without 5 endothelial and lymphocyte clusters*) to 14 major cell-types by inspecting the expression of marker genes and the correlation of the expression profiles between the clusters (Fig. 2a–c; *complete hierarchical annotation tree in* Supplementary. Fig. 4; *hierarchical clustering dendrograms by gene expression, by chromatin accessibility, and based on both modalities in* Supplementary. Fig. 5–7; *list of major-cell-type markers in* Supplementary. Data 4, 5). Neuronal cell clusters were organized in 10 major cell types (*6 inhibitory & 4 excitatory*) and glial cells clusters in 4 major cell types corresponding to the four major glial cell types: astrocytes (*7 cell clusters*), microglia (*one cell cluster*), oligodendrocytes (*one cell cluster*) and oligodendrocyte precursor cells (*'OPC', 5 cell clusters*). The glutamatergic/excitatory neuron types were characterized by the expression of the markers *LINC00507* (4 cell clusters), *RORB* (11 cell clusters), *THEMIS* (3 cell clusters) and *FEZF2* (5 cell clusters). As expected, the large, transcriptionally active nuclei of the excitatory neurons were associated with a higher number of detected features (*both genes and chromatin accessibility peaks*) and number of reads per feature (*UMIs for gene expression and fragments for chromatin accessibility*). The GABAergic/inhibitory interneurons were characterized by the expression of the markers *PVALB, SST, VIP, TAFA1, LAMP5* and *PAX6* (*6, 7, 9, 5, 5 and 1 cell clusters, respectively*). Correlation analysis of the gene expression profiles in the 14 major cell types demonstrates high correlation of the expression across all genes and samples inside three major groups: inhibitory neurons, excitatory neurons and glial cells (Fig. 2a, b), with some significant correlation of the expression profile of *FEZF2*-excitatory neurons also with inhibitory neurons. Hierarchal clustering of the major cell types in the reduced latent space spanned by the 50 top latents from chromatin accessibility and 50 top

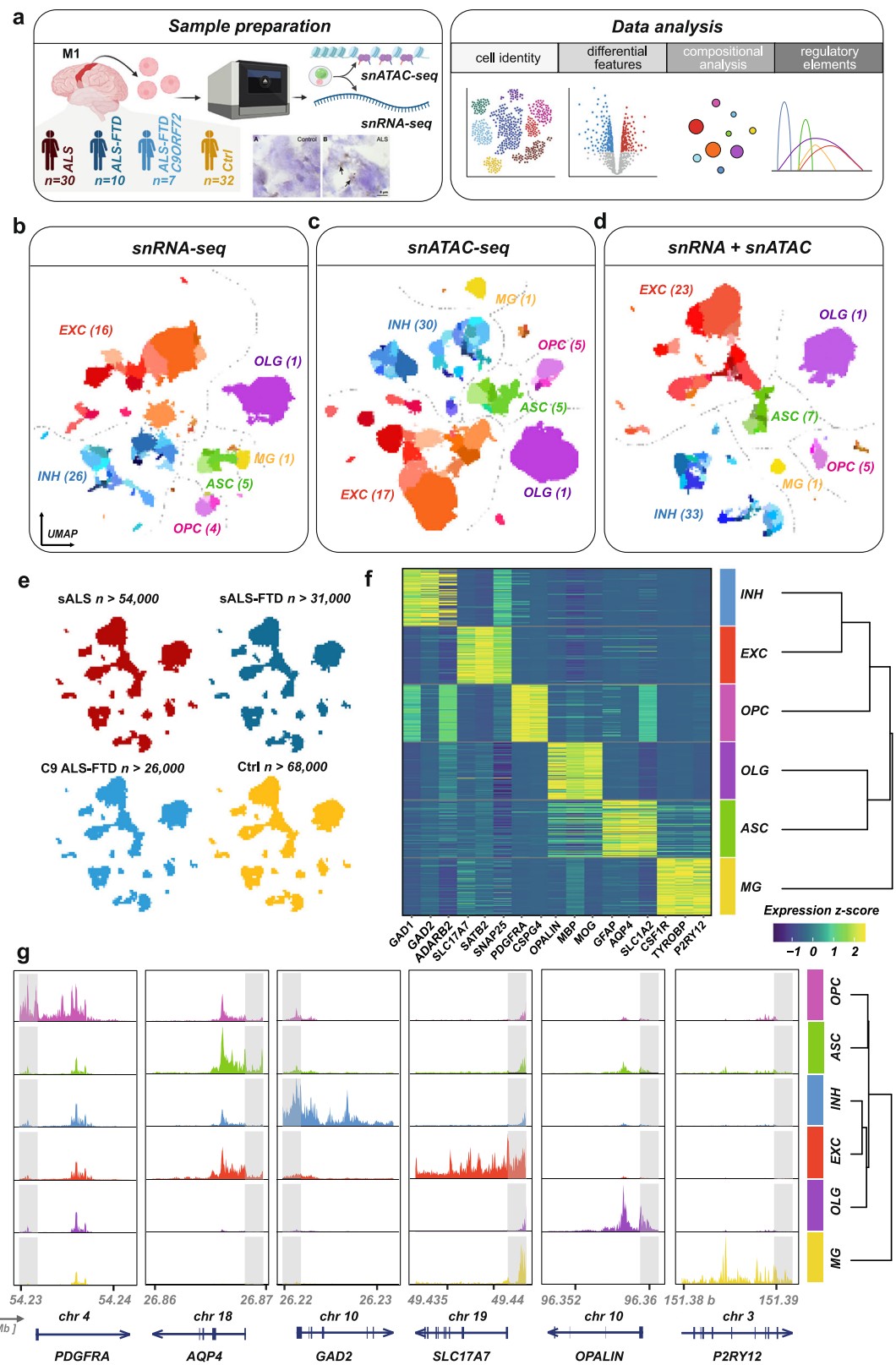

latents from gene expression demonstrated that the association of the major cell types is also preserved when considering the chromatin accessibility profiles (Fig. 2a). Similarly, the major cell type associations were similar when considering the whole gene expression data (Fig. 2c, *10,000 randomly sampled nuclei*). All 14 major cell types were well-represented in the three disease groups, both ALS-FTD types (*familial with C9ORF72 HRE and sporadic*), both sexes and all batch and cohort

groups (Fig. 2d), with a significant variation of numbers between donors (Supplementary. Fig. 8 *and* Supplementary. Data 22–24). *RORB-* and *LINC00507-*excitatory neurons and oligodendrocytes were the most abundant cell types, representing ~ two thirds of the entire dataset (*64% together; RORB: 16%, LINC00507: 21%, oligodendrocytes: 26%*) (Supplementary. Data 6). Of note, neuronal cell clusters, especially inhibitory, were more diverse in their expression profiles,

**Fig. 1 | A multi-omic single-nucleus dataset of the human motor cortex to study ALS/ALS-FTD. a** Schematic summary of the study design. We sampled nuclei from the primary motor cortex (M1) of 30 sporadic ALS (sALS) patients, 10 sALS-FTD patients, 7 familial ALS (fALS)-FTD patients (*C9ORF72 HRE*) and 32 non-affected controls ('Ctrl'). Diagnosis was pathologically confirmed by the presence of phospho-TDP-43 neuronal inclusions in all disease cases. Chromatin accessibility and transcriptome were assayed from the same nucleus with ATAC-Seq and 3'GEX simultaneously. **b–d** Integrated multi-modal clustering reveals fine-resolved clusters of cellular identity. Illustrative UMAP plots with coloring of distinct clusters based on gene expression (**b**, *n = 58* clusters), chromatin accessibility (**c**, *n = 64* clusters) or both (**d**, *n = 75* clusters, weighted nearest-neighbor, *WNN*). For distances and relations between clusters, see dendrograms in (**f, g**). Major cell types with number of subclusters in parentheses; *EXC*: excitatory neurons, *INH*: inhibitory neurons, *ASC*: astrocytes, *OLG*: oligodendrocytes, *OPC*: oligodendrocyte precursor cells, *MG*: microglia; vascular and immune cells categorized as "Other cells" in brown (no labels). Dotted lines: arbitrary cell domain borders. **e** UMAP plots by disease group with the number of nuclei from each group. Major cell types and their subclusters were comparably represented in each case group. **f** Gene expression markers of the major cell types identified with WNN, heatmap of expression z-scores. Each cell-type column represents 100 randomly sampled metacells constructed from 100 k-nearest neighbors within this cell type. Dendrogram: hierarchical clustering of the major cell types based on all cells and the transcriptomic ambient space. **g** Chromatin accessibility markers of the major cell types identified with WNN. The normalized signal plotted on the y-axis is scaled identically in each panel for a genomic region. Genomic regions 2 Kbp upstream of the promoter of canonical transcripts shaded in gray (*PDGFRA*: ENST00000257290, *AQP4*: ENST00000383168, *GAD2* ENST00000376261, *SLC17A7*: ENST00000221485, *OPALIN*: ENST00000371172, *P2RY12*: ENST00000273432); x-axis: genomic coordinates (GRCh38) in 10 Mbp steps. Dendrogram: hierarchical clustering of the major cell types based on all cells and the chromatin accessibility ambient space. Icon art in (**a**) (left) generated with BioRender.com (*Created in BioRender. Lee, J. (2026)* https://BioRender.com/05r88gx).

resulting in a higher number of smaller cell clusters than in glial cells at the same clustering resolution. For example, all oligodendrocytes clustered in a single large, homogenous cell cluster when clustering was performed at the same community resolution, together with all other nuclei in the dataset. Due to their comparably low numbers and highly correlated expression profiles, we combined *VIP*- and *TAFA1*-inhibitory neurons ('Inh VIP/TAFA1') and *LAMP5*- and *PAX6*-inhibitory neurons ('Inh LAMP5/PAX6') to facilitate further analysis. Our dataset reached an increased cell-type resolution compared to previous studies. It was comparable to previous reports regarding sequencing depth (*UMIs per nucleus, genes detected per nucleus*), number of nuclei per cluster, and cell-type proportions, but presented with a lower number of nuclei per sample (Supplementary Fig. 9). Furthermore, the set of detected (*expressed*) genes and the magnitude of expression were highly correlated to previous studies, as well as the KEGG terms enriched in this 'background' gene list (Supplementary Fig. 10). Remarkably, the segmentation of nuclei to different well-defined cell identities was highly concordant between our study and the studies of Pineda et al.[20], Li et al.[21] and Gittings et al.[18] (Supplementary Fig. 9k–n), demonstrating the robustness and reproducibility of cell-type identification.

Similarly to previous reports with single-nucleus sequencing in ALS/ALS-FTD[17,18,20,21], we did not observe a general change of a specific cell type in ALS, ALS-FTD or C9ORF72-ALS-FTD (Supplementary Fig. 11 and Supplementary Fig. 21). To ensure no technical bias, we analyzed cell-type proportions with three statistical frameworks: deep-learning based Bayesian compositional analysis with scCODA, hierarchical with tascCODA, and statistical modeling by mixed linear models with MASC. We also excluded bias by samples with lower total cell numbers by repeating the MASC analysis only with samples with a high number of nuclei. Still, we observed a comparable proportion of cell clusters, cell types, major cell types and cell classes, except for increased astrocytes and Exc RORB neurons (*in ALS and ALS-FTD resp.*) and reduced LAMP5 Inh neurons and microglia (*in ALS and ALS-FTD resp*) with MASC, which could not be confirmed with the other methods. However, we observed some stratification, with decreased neuron numbers (*e.g., Exc LINC00507 neurons*) in single samples, pointing to heterogeneity due to underlying phenotypical variation.

## Spatial transcriptomics allow for the mapping of the transcriptional cell types to their histological counterparts

Next, we sought to map the defined major transcriptional cell types to their histological correlates in the human cortex. ALS-FTD affects specifically the motor and the DLPF cortex in a progressive manner, which is thought to reflect the spread of pathology[7]. These areas are characterized by histologically defined morphological layers 'L1'-'L6' rich in neuronal somata of interconnected glutamatergic and GABAergic neurons, with pathology progressing from deep to upper cortical layers. The large pyramidal neurons of the deep layer L5b in the motor and the DLPF cortex are the only extratelencephalic neurons and traditionally considered to be the selectively vulnerable cells in ALS/ALS-FTD ('Betz cells' in the motor cortex). Nonetheless, TDP-43 pathology is found also in cells of other layers, mostly superficial and deep layers including L2, L3 and L6[7]. For the histological mapping of the major cell types, we utilized the publicly available LIBD spatial transcriptomic dataset of the adult human DLPF cortex, consisting of 12 slides from 3 neurologically unaffected adult cases and well-annotated cortical cell layers[22]. We built a cell-type characterizing gene expression signature for all major cell types from our multi-omic dataset and quantified it in the spatial transcriptomic dataset (Fig. 3) by gene score expression and by spot deconvolution of cell types (*numbers and lists of cell-type markers in* Supplementary Data 4 and Supplementary Data 5, *negative control with gene signatures composed of randomly sampled genes in* Supplementary. Fig. 12). We found that the excitatory neuronal types were found across all cortical layers, with Exc LINC00507 cells predominantly in L1 and L2/3 layers (*hereafter 'L2-3 intratelencephalic neurons'*), Exc RORB cells in L3-5 (*hereafter 'L3-5 intratelencephalic neurons'*), and Exc THEMIS and Exc FEZF2 cells in L5/6 (hereafter 'L6 intratelencephalic' and "L5-6 Near-projecting, extratelencephalic and corticothalamic' neurons, respectively) (Fig. 3a–d). Inhibitory neurons were more diffusely distributed across all layers, with Inh LAMP5/PAX6, Inh TAFA1/VIP and Inh SST cells predominantly in the upper cortical layers L1-3 and Inh PVALB neurons in all cortical layers (Fig. 3e) (hereafter referred to as 'interneurons' with their markers). As expected, glial cells were concentrated in the underlying white matter (*all types*), as well as L1 (OPCs, astrocytes and microglia) (Fig. 3e). Surprisingly, we observed an apparent enrichment of microglia in L1 in this highly validated dataset, but we could not identify whether this is an absolute enrichment of microglia or a relative enrichment due to lower numbers of other cell types without further studies. To identify the transcriptional equivalent of the large pyramidal cells of layer Vb of the motor cortex, we calculated a score based on 410 gene expression markers of extratelencephalic neurons[19] (Supplementary. Data 5). Interestingly, the signature was enriched very strongly and specifically only in one cell type, Exc FEZF2-NTNG1 cells, which we identified as the neuronal community hosting the Betz cells of the motor cortex. Correspondingly, the Exc FEZF2-NTNG1 signature was found in very rare, single spots with very high signature expression in the spatial transcriptomic data, as expected for large but rare cells (Fig. 3f and Supplementary Fig. 13). Furthermore, the Exc FEZF2-NTNG1 cells overlapped with very high accuracy with the VAT1L neurons from Pineda et al.[20], the L5 extratelencephalic neurons from Gittings et al.[18] and the Exc L5 ET FEZF2-ADRA1A neurons from Li et al.[21], unequivocally confirming their identity (Supplementary Fig. 9n).

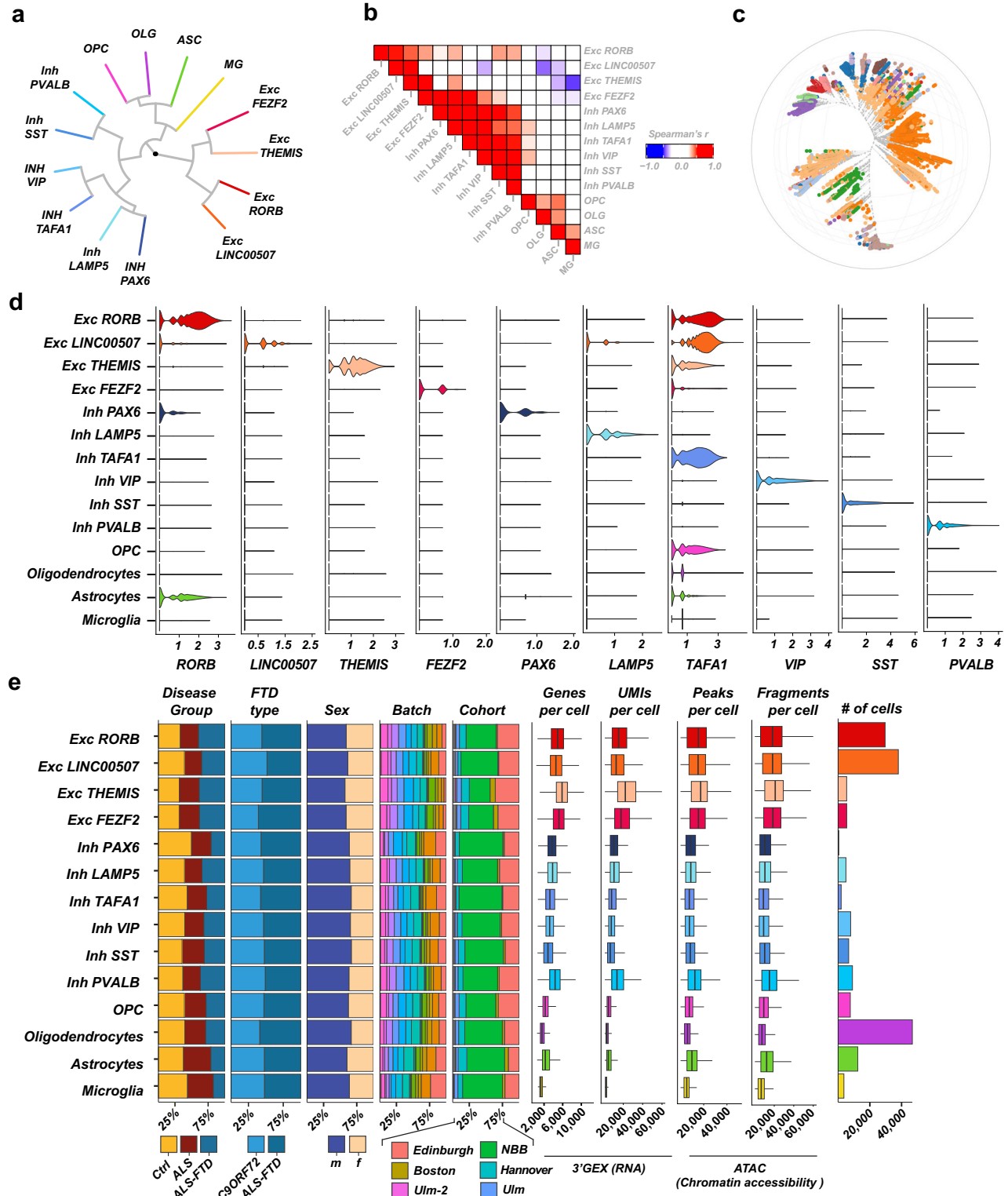

**Fig. 2 | Hierarchical cell-type annotation of the human motor cortex dataset.**
**a** Major cell types identified in the human motor cortex dataset. Hierarchical clustering (average linkage, Euclidean distance) of the 14 major cell types over the top 50 latents from each modality (gene expression & chromatin accessibility).
**b** Heatmap of the pairwise correlation between the gene expression profiles of each major cell type. Scale: Spearman's ρ. **c** Bonsai representation of 10,000 randomly sampled nuclei with the major cell type annotated (colors not equivalent to colors in **a** and **d**). **d** Expression of markers used to annotate a major cell type against the other major cell types in the same cell class (as log-normalized counts). Violin plots from n = 180016 nuclei. **e** Characteristics of each major cell type: proportion of cells

belonging to each disease group (%), proportion of ALS-FTD cells belonging to the *C9ORF72* group (%), proportion belonging to each sex (%), proportion of cell originating from each technical batch (%), proportion of cells originating from each cohort (%), number of genes detected per cell, number of UMIs per cell, number of peaks detected per cell, number of fragments per cell, and total number of cells in each major cell type. Bars of percentages totaling to 100%. Box plots with median (center line), first and third quartiles (25/75%) and minimum/maximum value within 1.5 x interquartile range from the first/third quantile, respective (lower/upper whisker). Data from n = 180016 nuclei. Source data are provided as a Source Data file.

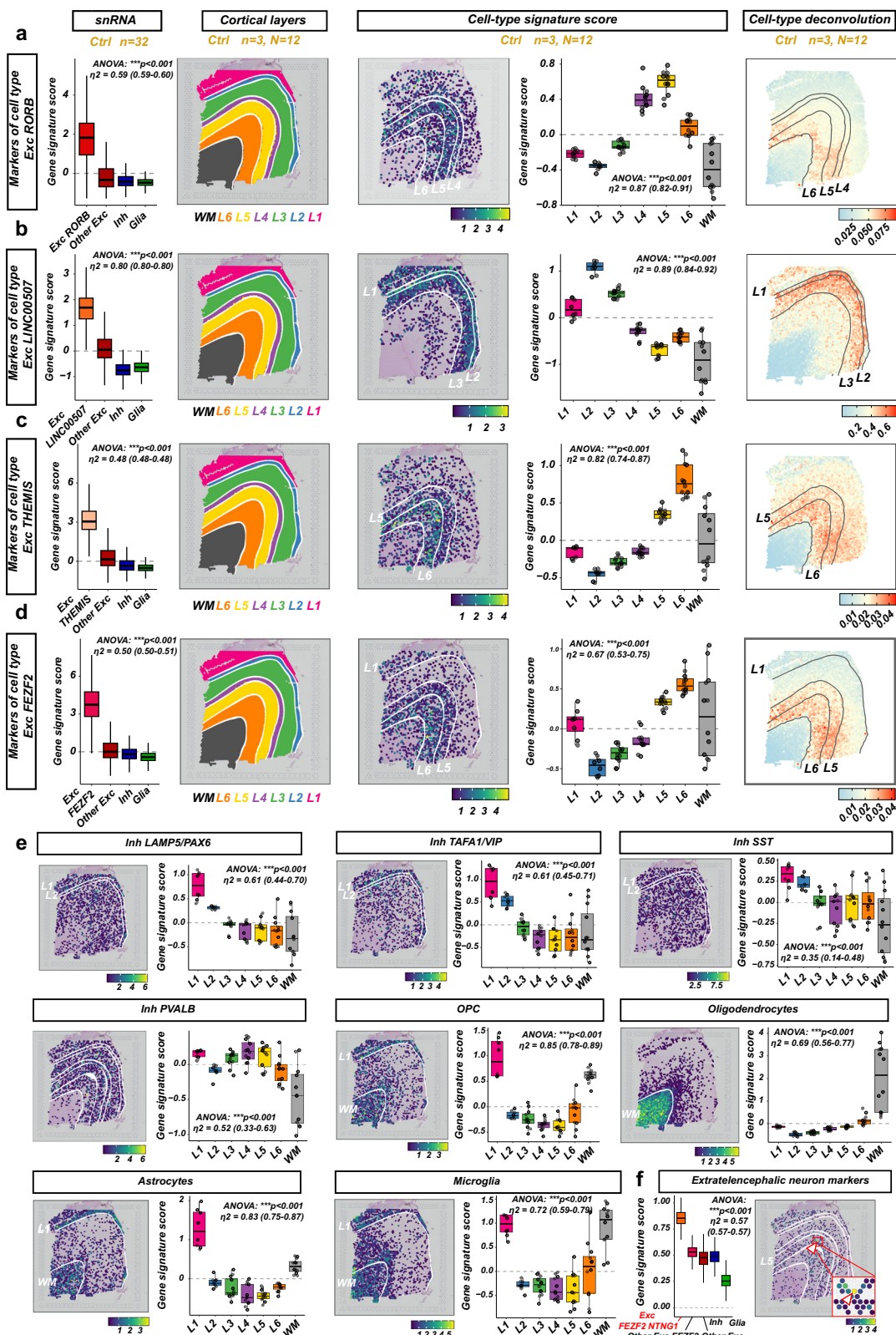

## The ALS/ALS-FTD motor cortex is transcriptionally altered across hierarchical cell types

We next investigated in detail the transcriptional changes in ALS and ALS-FTD motor cortex. The cell-type unaware analysis (*i.e., all cells from a donor are combined to a single sample*) confirmed robust transcriptional changes in ALS and ALS-FTD (Supplementary Fig. 14). We employed surrogate variable analysis (*SVA*) to account for hidden batch variables like post-mortem interval (*PMI*) and onset site, as due to incomplete data, we were unable to directly include these variables as covariates in the differential gene expression (*DGE*) analysis. Combining DGE analysis with SVA strongly increased the sensitivity of the analysis while maintaining the specificity of the results, as demonstrated by systematically increasing the number of covariates in the analysis (Supplementary Fig. 14). We identified 667 differentially

**Fig. 3 | Mapping of transcriptomic cell types to histological correlates with spatial transcriptomics.** The cell-type gene signature for each major cell type was calculated against all other cells in all neurologically unaffected samples ('Ctrl', $n = 32$) and then quantified in each spot of the publicly available LIBD human dorsolateral pre-frontal cortex dataset. **a–d** Characterization of the excitatory neuronal major cell types. Boxplots (left) show the enrichment of the calculated cell-type signature in the respective major cell types (cells as observation points) as an indication of how strong/specific is the cell-type gene signature; cluster plots show the annotation of the spot to the human cortical layers as provided by LIBD, WM: white matter, L1-L6: layers 1-6; spatial gene expression plots show the scaled, cumulative expression of the respective cell-type signature ('score'); boxplots (right) show the enrichment of the calculated cell-type signature in each cortical layer (observation points: slides; $n = 3$ samples, 4 slides per sample); spatial heat-map plots: heatmap of major-cell type deconvolution in each spatial transcriptomics spot (scaled percentage over all spots, higher scale values mean cells of the respective type are more concentrated in distinct spots). **e** Characterization of the inhibitory and glial cell types. Spatial gene expression plots show the scaled, cumulative expression of the respective cell-type signature ('score', same as **a–d**). **f** Identification of extratelencephalic cells. Gene signature score was calculated from the expression of 406 expressed extratelencephalic cell markers. Exc FEZF2-NTNG1 cells show a specific high enrichment of the score. Single, dispersed cells with very high scaled-score enrichment (> 3) can be found throughout Layer 5 (red arrowhead, inlet: enlarged area from box in red). Boxplot and spatial expression plot as in (**a–d**). The same slide with a good topological distinction of all layers (sample id '151674') was used for all spatial plots. Layers in which the respective cell types are enriched are annotated with contours (manually added) and names in white; the same contours used for all cluster/spatial expression slides. Boxplots show different number of samples per group because some slides did not have all layers detected. Box plots with median (center line), first and third quartiles (25/75%) and minimum/maximum value within 1.5 x interquartile range from the first/third quantile, respective (lower/upper whisker). ANOVA and effect sizes two-sided with 95% CI. Source data are provided as a Source Data file.

expressed genes ('DEGs') in ALS (absolute log2 fold-change > 0, fdr < 0.05; sample permutation control: 3 DEGs) and 3,827 DEGs in ALS-FTD (sample permutation control: 19 DEGs) (Supplementary. Fig. 14, lists of DEGs in Supplementary. Data 7–9), even despite the lower number of samples (and thus decreased statistical power) in the ALS-FTD analysis (ALS/Ctrl: $n = 30/32$; ALS-FTD/Ctrl: $n = 17/32$). Thus, transcriptomic changes were stronger in ALS-FTD. Comparison of DEG numbers between C9ORF72-HRE positive and sporadic ALS-FTD cases revealed that the strong transcriptomic changes are found in both disease subgroups (Supplementary Fig. 15). Comparing the magnitude and direction of regulation (up/down) revealed a strong correlation of gene changes in ALS and ALS-FTD, with > 96% of the common DEGs changed concordantly ($n = 173/180$, Pearson's $r = 0.90$, ****$p < 0.0001$). ALS/ALS-FTD DEGs showed a large overlap with known KEGG pathways of ALS (Supplementary Fig. 14). DEGs were enriched in pathways associated with aerobic metabolism ('oxidative phosphorylation', 'metabolic pathways') and neurodegenerative diseases ('ALS', 'Parkinson's disease', 'Huntington's disease', 'prion disease'), probably driven by the high overlap of the genes annotated to these pathways with the oxidative phosphorylation pathways. DEGs included known targets of TDP-43 pathology, including STMN2, EPHA4, NPTX2 and its receptor NPTXR, which were all down-regulated in ALS/ALS-FTD. Taken together, this low-resolution analysis demonstrates robust transcriptomic remodeling in ALS/ALS-FTD, with strong overlap between both disease groups, thus confirming its functional relevance.

To assess the contribution of different cell types, we next investigated differential expression events across the 12 major cell types. Increasing the resolution increased the power to detect DEGs and demonstrated transcriptomic changes in specific major cell types (Fig. 4). Transcriptomic changes in ALS were almost exclusively in excitatory neurons, with some DEGs also in VIP and Parvalbumin interneurons and in astrocytes, while in ALS-FTD, more DEGs were detected in all major cell types (Fig. 4a–d). In general, transcriptional dysregulation was balanced between up- and down-regulated genes, with slightly more down-regulated genes in most cell types. In both disease groups, the highest number of DEGs was detected in L2-3 intratelencephalic ('L2-3 ITC') and L3-5 intratelencephalic ('L3-5 ITC') excitatory neurons. The number of detected DEGs strongly correlated with the total number of reads per major cell type, demonstrating that direct comparison of the strength of transcriptomic dysregulation between the major cell types is not feasible without equilibrating the statistical power (mostly for read numbers) (Fig. 4a–d). At the same time, it suggests some specificity of the differences between major cell types; for example, a relatively high number of DEGs were detected in oligodendrocytes in ALS-FTD, despite the comparable library size to Parvalbumin interneurons. Similarly, in both ALS and ALS-FTD, more DEGs were detected in L3-5 ITC excitatory neurons than in L2-3 ITC excitatory neurons, despite their lower numbers and smaller total library sizes (Fig. 4c, d).

Across the 11 L3-5 ITC excitatory neuron cell types, we found the most DEGs in RORB ERRB4 and RORB LNX2 cells in ALS, and in RORB CUX2 and RORB LNX2 cells in ALS-FTD. From the 4 L2-3 ITC excitatory cell types, we found the most DEGs in the most abundant cell type, LINC00507 FREM3 both in ALS and ALS-FTD (Fig. 4e–l). DEGs found in ALS and ALS-FTD were strongly expressed in layers L2-6 and less in L1 and the white matter (Fig. 4m–o; 1,197/1,315 genes from ALS signature expressed in the spatial data, 3,851/4,666 from the ALS-FTD signature) in the neurologically unaffected. Together, we identified 5,471 genes which were differentially expressed in at least one major cell type in at least one condition (Fig. 4m and Supplementary Data 10). 510 of these genes were detected both in ALS and ALS-FTD (Supplementary Data 11). Many of the ALS/ALS-FTD genes were similarly regulated in major cell types also when statistical significance was not reached (i.e., fdr > 0.05, Supplementary Fig. 16). Among the signatures were genes associated with familial ALS/ALS-FTD: SOD1, TARDBP, TBK1, ALS2, BSCL2, DCTN1, GLE1, GRN, HNRNPA1, NEFH, PFN1, SETX, UBQLN2, VAPB, VEGFA and with known association to ALS/ALS-FTD: NEFL, STMN2, UNC13A, NPTX1, NPTX2 and NPTXR, amongst others.

Interestingly, gene expression was regulated more concordantly between different cell types in a disease group than between disease groups for the same cell type (Supplementary Fig. 17). These findings are highly concordant with Pineda et al.[20] and confirm that the robust transcriptomic changes in major cell types are more 'disease-specific' than 'cell-type specific'. Altogether, transcriptomic changes were highly convergent both across cell types and across disease groups.

In summary, transcriptomic analysis across hierarchical cell types in ALS/ALS-FTD reveals predominantly concordant changes across cell types and disease genotypes/phenotypes, indicating common underlying factors. Thus, we next explored the roles of epigenetic remodeling and TDP-43 pathology.

**Chromatin remodeling is associated with a subset of the ALS/ALS-FTD transcriptional changes**

Next, we investigated the hypothesis that at least a part of the transcriptomic signature of ALS/ALS-FTD is associated with underlying changes in chromatin accessibility as a cumulative proxy measure for epigenetic remodeling. We exploited the simultaneous profiling of chromatin accessibility and gene expression in the same nucleus across all cells in the dataset to detect significant correlations ('peak links') between chromatin accessibility regions ('peaks') and expression of nearby genes, which are more difficult to detect in designs with less observations (i.e., on a sample level or not measured in the same cell) (Fig. 5a). In total, we detected > 53,000 significant associations between 36,721 unique peaks (~ 18% of all peaks) and > 6000 genes (~ 18% of all expressed genes), with most genes associated to < 20, but up to 68 peaks, and most peaks associated to one gene or a maximum of 12 genes across all peaks (Fig. 5b). Peaks associated with gene

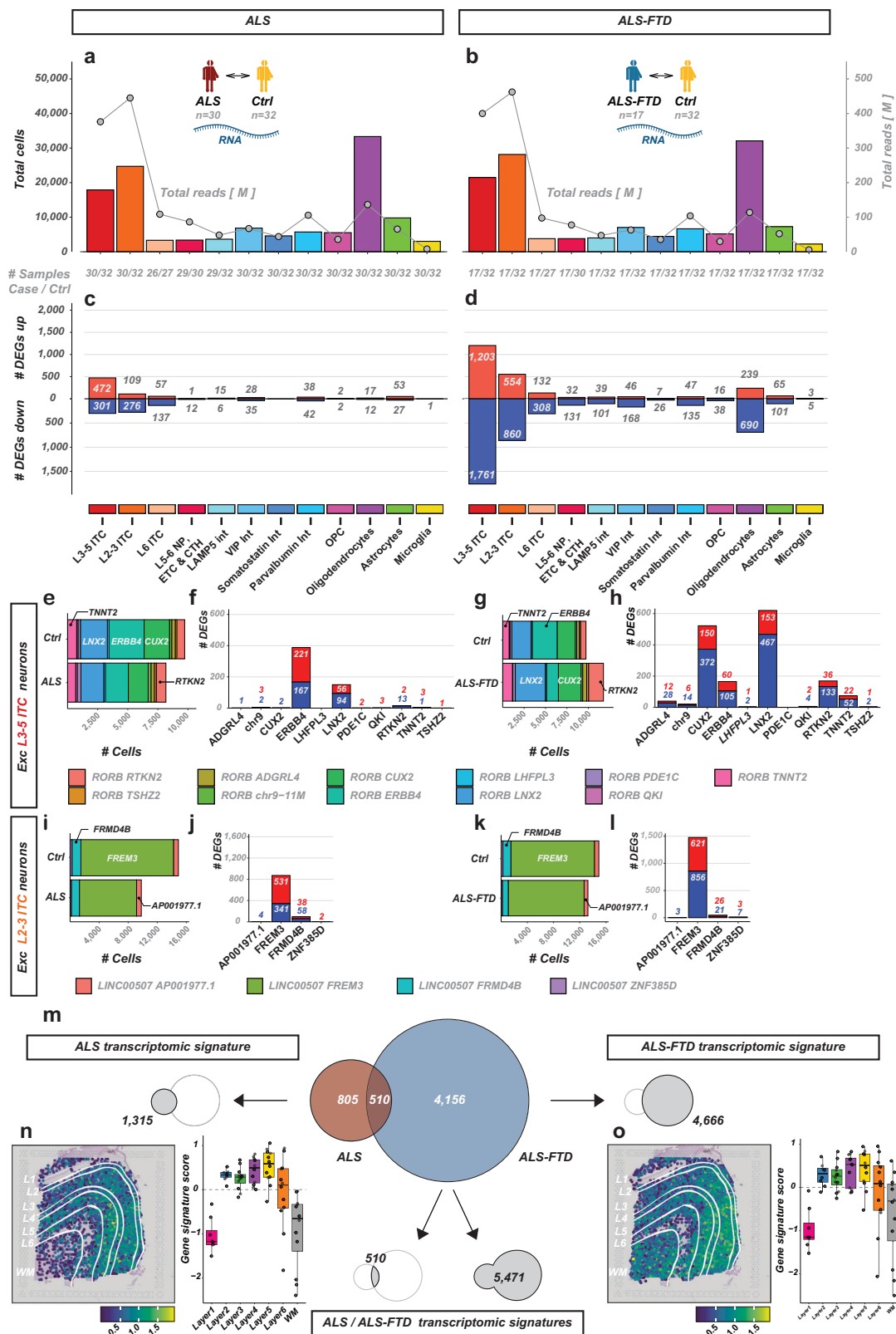

expression were enriched in promoter elements (Fig. 5c). Of the 5,471 genes in the ALS/ALS-FTD transcriptomic signature, 2,271 (*42%*) were annotated a peak link, showing a > 4-fold enrichment of chromatin-associated peaks in the signature (*compared to a 1,000-fold random sampling from genes with comparable expression levels*). This suggests that ALS/ALS-FTD genes are specifically associated with chromatin accessibility.

Differential accessibility analysis identified differentially accessible peaks (*"differentially accessible regions, 'DARs'"*) in several major cell types: astrocytes (*n = 531/52 DARs in ALS/ALS-FTD resp.*), oligodendrocytes (*n = 46/24 DARs in ALS/ALS-FTD resp.*), excitatory L3-L5 ITC neurons (*n = 51/2,277 DARs in ALS/ALS-FTD resp.*), excitatory L6 ITC neurons (*n = 1/5 DARs in ALS/ALS-FTD resp.*) and excitatory L2-3 ITC neurons (*n = 33/2,973 DARs in ALS/ALS-FTD resp.*) (Fig. 5d, f). Similarly to

**Fig. 4 | Transcriptional changes in ALS/ALS-FTD across major motor cortical cell types.** Differential gene expression analysis with each donor sample as an observation point ('pseudo-bulk') in ALS (left) and ALS-FTD (right). **a, b** Number of cases, total cell numbers (bars) and total library size (all cases and controls, gray bulleted line) for each major cell type. Note the overrepresentation of L3-5 ITC and L2-3 ITC excitatory neurons and oligodendrocytes. **c, d** Number of DEGs detected for each comparison at fdr < 0.05 and | log2-fc > 0 | (up-regulated: red bars, down-regulated: blue bars). **e–l** Proportion of each cell type in the L3-5 ITC neuronal population (**e, g**) and the L2-3 ITC neuronal population (**i, k**), with the number of DEGs detected in each cell type (**f, h, j, l**). **m** Disease-specific transcriptomic

signatures defined from all DEGs detected in at least one major cell type in ALS and ALS-FTD. **n** Expression of the ALS and ALS-FTD (**o**) transcriptomic signature in the neurologically normal adult human cortex. Spatial maps with the cumulative, scaled signature expression score and boxplots with the enrichment of the signature in each cortical layer (observation points: slides; $n = 3$ samples, 4 slides per sample). Box plots with median (center line), first and third quartiles (25/75%) and minimum/maximum value within 1.5 x interquartile range from the first/third quantile, respectively (lower/upper whisker). Source data are provided as a Source Data file. Icon art in (**a, b**) generated with BioRender.com (Created in BioRender. Lee, J. (2026) https://BioRender.com/nfa1g1r).

the DEGs, DARs were often shared between cell types and their regulation correlated strongly between cell types where they were differentially accessible (Fig. 5e, g, h). The accessibility of significant DARs was strongly positively correlated to the expression of the associated genes across cell types and disease groups (Fig. 5i, *yellow dots*). To investigate whether chromatin accessibility peaks are associated with SNVs previously identified in GWAS studies, we compiled a list of 222 unique ALS GWAS SNVs from the GWASdb collected from different studies. In total, 24 SNVs overlapped with chromatin accessibility regions. Analysis of the combined chromatin accessibility of these positions revealed no increased/decreased accessibility in them, except for oligodendrocytes in ALS-FTD (Supplementary Fig. 18). Taken together, this data demonstrated that a portion of the transcriptomic changes in ALS/ALS-FTD motor cortex is associated with chromatin changes.

## Nuclear TDP-43 pathology selectively affects excitatory L2-3 intratelencephalic, L5 extratelencephalic and L6 intratelencephalic neurons

To determine which ALS/ALS-FTD DEGs were associated with TDP-43 pathology (*e.g., resulting from or leading to it*) in a cell-type specific manner, we aimed to define the transcriptomic signature of nuclear TDP-43 depletion. Of note, the highly cited previous report has described a TDP-43 pathology transcriptomic signature, but it was derived with bulk sequencing[10]. This type of analysis returns cell-type markers next to the specific DEGs, because only some cell types will be affected by TDP-43 pathology. Indeed, as the authors of this study themselves, we found a strong enrichment of cell-type markers in the bulk-sequencing derived TDP-43 pathology signature (Supplementary Fig. 19). However, it is crucial to distinguish specifically up-/down-regulated genes from enriched cell-type markers. Therefore, we performed fluorescence-activated nuclei sorting coupled with single-nuclei RNA-seq (*FANS-snRNA-seq*) based on the TDP-43 signal in neuronal nuclei (Fig. 6a and Supplementary Fig. 20). TDP-43 pathology in vulnerable neurons is characterized primarily by depletion from the nucleus, mostly accompanied by its phosphorylation and aggregation in the cytoplasm. Thus, neuronal nuclei with TDP-43 pathology can be identified by the decrease in TDP-43 signal. We selected isolated nuclei singlets from 10 ALS-FTD samples with confirmed presence of TDP-43 pathology by selecting nuclei with high relative signal of NeuN (*RBFOX3, a marker of neuronal nuclei; found also in some oligodendrocytes*, Fig. 6b) and sorted them into nuclei with TDP-43 pathology (TDP-43 Low) and without TDP-43 pathology (TDP-43 High). Around 30–40 % of all nuclei were positively selected (Fig. 6e; *no debris, singlets, DAPI +, NeuN +*), of which <6 % were TDP-43 Low (Fig. 6f). As expected, ~ 1 % of all neuronal nuclei were TDP-43 Low. We then employed the nuclei fractions for single-nucleus RNA-seq with the 10 × 3′GEX kit and acquired a total of 12,225 nuclei transcriptome profiles (*7189 TDP-43 High, 5036 TDP-43 Low*) after QC filtering. To identify the cell type of the nuclei affected by TDP-43 pathology, we transferred the well-defined cell-type labels from the high-resolution ALS-FTD multiome nuclei dataset to the FANS-seq nuclei based on the expression of anchor features with Seurat ("*Methods*"). Clustering and visualization with Bonsai[23] and UMAP

confirmed that despite the strong expected transcriptomic changes by TDP-43 pathology, nuclei still primarily clustered by cell type and thus the cell type can be identified (Fig. 6 h–l). More than 95% of the selected nuclei were neuronal, as intended (Fig. 6m). Even though the *RBFOX3* expression was lower in inhibitory than in excitatory neurons (Fig. 6b), we found that the proportions of excitatory ( ~ 75%) and inhibitory neurons ( ~ 25%) in the TDP-43 High data was comparable to the unsorted multi-omic dataset (Fig. 6n) and previous reports[17,24], confirming that the sorting strategy was appropriate and did not passively enrich excitatory neurons due to stronger *RBFOX3* expression. While all 8 neuronal cell types (*4 inhibitory, 4 excitatory*) and oligodendrocytes were detected in the TDP-43 High nuclei fraction, we detected almost exclusively excitatory neurons in the TDP-43 Low nuclei fraction ( > 95 %) (Fig. 6n). All major excitatory cell types were affected, but not equally represented, with an enrichment of L2-3 ITC and L5-6 ITC- and ETC neurons (Fig. 6o and Supplementary Data 25). Again, *TARDBP* expression was not lower in L2-3 intratelencephalic neurons (Fig. 6c, d), suggesting that the expansion of this cell type in the TDP-43 Low fraction is specific. Similarly, Somatostatin interneurons were the most represented interneurons in TDP-43 Low nuclei, despite their relatively high TDP-43 expression (Fig. 6d), reiterating the specificity of the TDP-43-based sorting. Thus, nuclear TDP-43 depletion in the motor cortex is highly selective for specific neuronal cell types.

To definitely confirm that nuclei in the TDP-43 Low fraction are nuclei from cells affected by TDP-43 pathology, we sought to reproduce known features of TDP-43 pathology: previously described gene signatures, cryptic exon (*CE*) inclusion and alternative poly-adenylation. Cell-type unaware ("*pseudo-bulk*") DGE analysis revealed a robust dysregulation of gene expression in TDP-43 Low nuclei compared to TDP-43 High nuclei, with >1600 differentially expressed genes (Fig. 7a and Supplementary Data 12). More than 40% of the DEGs were amongst the DEGs previously described in bulk-RNA-seq TDP-43 FANS in the frontal cortex (*Liu et al.*)[10] and in single-nucleus RNA-seq of nuclei predicted to be affected by TDP-43 pathology (*Gittings et al.*)[18] (Fig. 7 b). The dysregulation of the genes found in common with these studies was highly concordant both in direction (*up- vs. down-regulated*) and magnitude (Fig. 7c, d). Thus, transcriptomic signatures confirmed TDP-43 pathology in the TDP-43 Low fraction. The known and recently well-described targets of TDP-43 pathology, *STMN2* and *NPTX2* were also down-regulated in TDP-43 Low nuclei, in accordance with previous reports. Interestingly, *NPTXR*, the receptor for *NPTX2* was also down-regulated, in line with previous findings with iCLIP data that *NPTXR* splicing is regulated by TDP-43[25]. In addition, we analyzed CE inclusion in TDP-43 Low nuclei. Of the 66 previously described alternative splicing events in TDP-43 pathology (*Ma et al.*)[14] with bulk RNA-seq, two could be previously detected in 10X Genomics 3′GEX single-nucleus RNA-seq data: *STMN2* and *KALRN* CEs[18]. We could detect reads spanning the CE junctions in both genes with significantly higher counts than the previous report, but only in the TDP-43 Low nuclei (Fig. 7e–g). Differential poly-adenylation usage analysis also demonstrated >1000 differentially used distal poly-adenylation sites. Differential poly-adenylation was visible e.g., at the previously described genes *MARK3* and *ELP1*[15,26,27] (Fig. 7h–j). Thus, we could confirm with high confidence TDP-43 pathology in the TDP-43 Low nuclei.

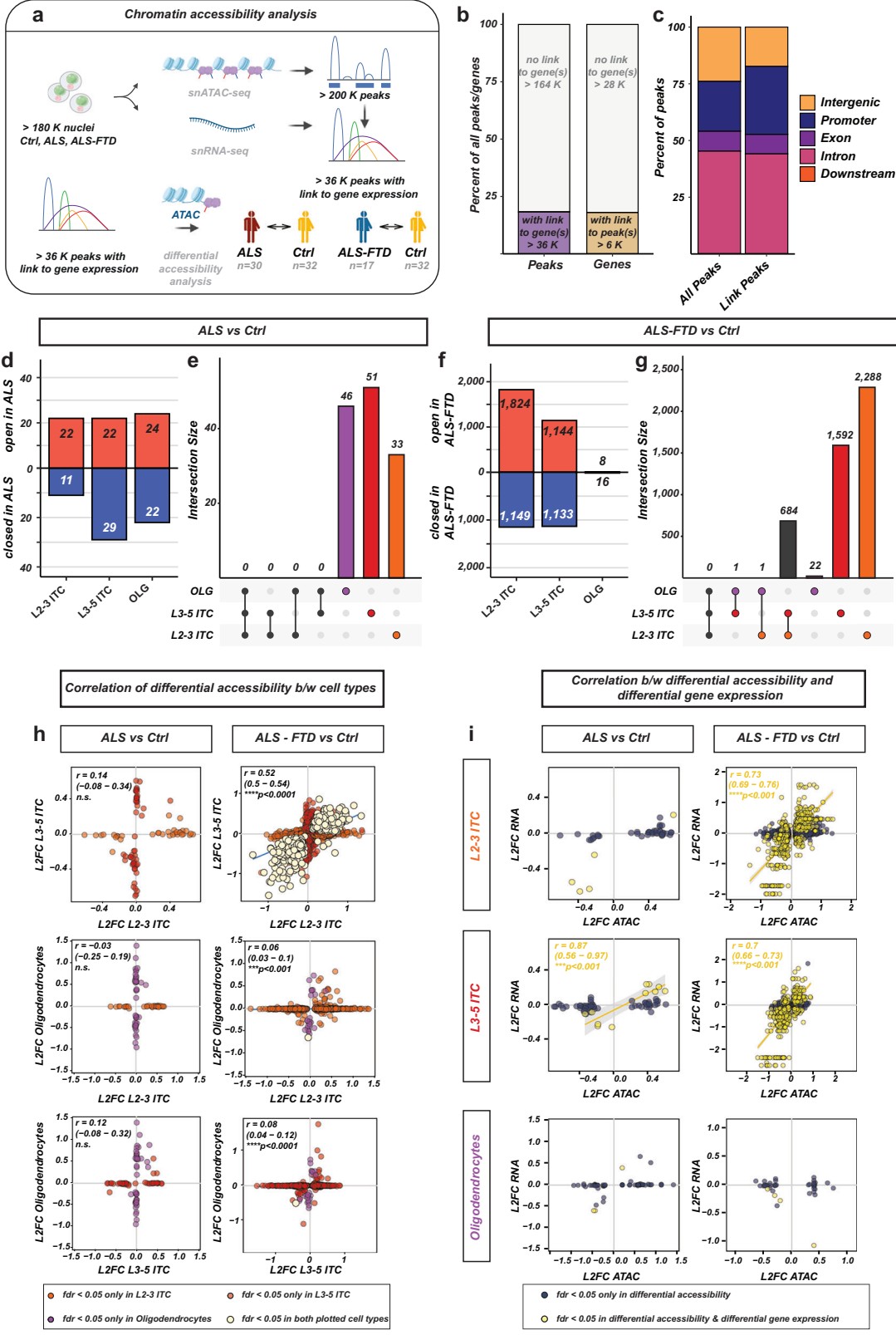

## Transcriptional effects of TDP-43 pathology are cell-type specific

Next, we asked whether the transcriptional effects of TDP-43 pathology are cell-type specific, as suggested by previous studies[28]. Differential gene expression analysis with DESeq2 and pseudo-bulk in each cell type revealed variable DGE numbers in the four major excitatory cell types (Fig. 8a and Supplementary Data 13–16). The highest number

of DGEs was found in L5/L6 NP, ETL and CTH neurons (*FEZF2* +), despite their lower numbers and library sizes (and thus decreased statistical power) (Fig. 8d). Similarly, CE inclusion was cell-type specific, with most counts of CE junctions in L2-3 ITC neurons. However, compared to the total number of cells and counts, L5-6 NP, ETL, and CTH, and L6 ITC neurons had the highest CE counts (Fig. 8g, h). Interestingly, *STMN2* and *KALRN* CEs were cell-type specific, with more

**Fig. 5 | Chromatin accessibility changes in ALS/ALS-FTD across the major cortical cell types.** The simultaneous profiling of chromatin accessibility (ATAC-seq) and gene expression (RNA-seq) allowed for the identification of > 36 K genomic regions whose accessibility was directly linked to gene expression (**a**). The differential accessibility of these regions between ALS/ALS-FTD and controls was analyzed across the 12 major cell types with each donor as a sample ("pseudo-bulk"). **b** Proportions of peaks (left) and genes (right) that were associated with one of the identified regulatory elements. **c** Annotation of the genomic regions in respect to genes in order of priority: 'Promoter': within 3,000 bp of the TSS; 'Exon'/'Intron': located in an exon/intron of a gene (5'UTR and 3'UTR recognized in exons); 'Downstream': downstream of the gene body'; 'Intergenic': significant distance to any nearby gene. "All Peaks": initial set of > 200 K peaks (after filtering), 'Link Peaks': > 36 K peaks associated with gene expression. **d–g** Number in differentially accessible peaks in the three major cell types with the strongest transcriptional changes

(**d**, **f**, pseudo-bulk with SVA, 8 SVs). **e**, **g** overlap of the differentially accessible peaks between the three cell types (number of peaks above bars). **h** Correlation of peaks' accessibility in pairwise comparisons of the three major cell types in ALS (left) and ALS-FTD (right). Each dot is a peak, peaks that were significantly changed in only one of the compared cell types in the respective cell-type color, peaks which were significantly changed in both cell types in yellow. **i** Correlation of peaks' accessibility with gene expression in each of the three cell types (rows) in both disease groups (columns). Only significant differentially accessible peaks plotted (even if not significant in gene expression). Correlation with the peaks for which the respective gene was also significantly differentially expressed (in yellow). Spearman's correlation (**h**, **i**) r with 95% CI and two-sided test p-value. The error band represents the 95% CI of the fitted regression line. Source data are provided as a Source Data file. Icon art in **a** generated with BioRender.com (Created in BioRender. Lee, J. (2026) https://BioRender.com/phqhvt1).

*STMN2* CE counts in L5-6 NP, ETL & CTH neurons and more *KALRN* CE counts in L6 ITC neurons (Fig. 8b). Compositional analysis of cell subtypes in each major cell type revealed the specific cell subtypes most affected: L2-3 *LINCO0507 + FREM3* + ITC, L3-L5 *RORB + ADGRL4+* and *RORB + LNX2* + ITC, L5 ETC (*FEZF2 + NTNG1 +* ) and L6 *THEMIS + LINCO0343* + ITC neurons (Fig. 8c). Despite the relatively large number of DEGs detected in each major cell type, most DEGs were specific to one of the cell types (Fig. 8e, f), concordant with an enrichment of cell-subtype markers due to the different proportions of cell subtypes. Nevertheless, some DEGs that were found in more than one major cell type were dysregulated concordantly (Fig. 8f), with notable exceptions in L5-6 NP, ETL & CTH neurons. In line with this, hierarchical clustering based on the DGE changes demonstrated that transcriptomic changes are more distinct in L5-6 NP, ETL & CTH neurons (Fig. 8f). DGE analysis with MAST showed a high number of DEGs in all five most affected cell types (Supplementary Data 17–21). In addition, we observed that the counts of CE reads were mostly found in these 5 cell types, further defining them as the transcriptional cell types affected by TDP-43 pathology.

### Cell-type specific transcriptomic changes in the ALS/ALS-FTD motor cortex are associated with chromatin changes and distinct from the TDP-43 transcriptomic changes

Next, we investigated how TDP-43-pathology-associated transcriptional changes relate to the general transcriptomic changes we observed in the ALS-FTD motor cortex. Hierarchical clustering of the log2-fold changes for each cell type and all genes significant in at least one comparison from the ALS-FTD vs. Ctrl and the TDP-43 $^{Low}$ vs. TDP-43 $^{High}$ nuclei demonstrated that these are two distinct sets of transcriptomic changes (Fig. 9a). Still, a significant proportion (n = 373, ****p < 0.0001 OR 2.92, 95% CI 2.54-3.34) of the TDP-43 $^{Low}$ DEGs were found also in the ALS/ALS-FTD DEGs, with a significant enrichment of a sole KEGG term "Axon guidance" (Fig. 9b). On a major excitatory neuron type level, overlap of TDP-43 $^{Low}$ DEGs with the ALS-FTD DEGs was variable but significantly enriched for all four major cell types (Fig. 9c), with little overlap to changes associated with chromatin accessibility. Among the chromatin accessibility changes, several signaling pathways were enriched uniquely for dysregulated genes associated with chromatin accessibility changes, but not with TDP-43 pathology. Interestingly, the 'Axon guidance' KEGG term enrichment in TDP-43 DEGs was driven by 7 genes, including *EPHA3*, *EPHA4* and *EPHA6* that were also dysregulated on the level of chromatin accessibility.

## Discussion

In the present study, we identify multi-modal disease signatures in the major cell types of the primary motor cortex in ALS/ALS-FTD, with notably more-pronounced changes in ALS-FTD cases. We then identify the transcriptional cell types affected by TDP-43 pathology and show that the effects of TDP-43 pathology are cell-type specific. By

leveraging several techniques and single-nucleus sequencing, we dissect transcriptomic changes and show a cell-type differential association to epigenetics and TDP-43 pathology. The disease signature is detectable across all major cell types, but is particularly enriched in excitatory neurons, especially in L2-3 ITC and L3-5 ITC neurons. Further analysis reveals distinct components of the transcriptional signature: some that are associated with TDP-43 pathology, and such that are not associated with TDP-43 pathology, but with epigenetic changes in signaling pathways. These findings underscore specific cell-type susceptibilities and suggest more severe molecular disruptions in ALS-FTD, pointing to potential targets for tailored therapeutic strategies.

The study of transcriptomic changes through RNA-seq has established itself as a powerful tool in the discovery of putative disease mechanisms and therapeutic targets. However, such efforts are often hindered in complex settings by the complexity of concurrent processes altering the transcriptome. For example, in ALS/ALS-FTD, a study of the disease transcriptome in the CNS is typically only possible as a cross-sectional snapshot at *postmortem*. Such a transcriptomic picture is the combined product of amongst others: putative transcriptomic changes that occur prior to disease and represent the initial pathologic changes; transcriptomic changes associated with the degenerative process, e.g., with TDP-43 aggregation and mislocalization; transcriptomic changes as a consequence of the degenerative process, as well as potential compensatory changes; finally, of transcriptomic changes that are incurred by the coordinated cell-cell communication, e.g., neuroinflammatory processes. Further complicating the picture, until the recent decade, microarray and RNA-seq studies were only possible in bulk, i.e., as a study of the mixture of many thousands of cells together. In such studies, transcriptomic changes in affected cells are not only obscured in the cell mixture, but also emerge as apparent changes from differences in the cell-type composition of the mixture. A step forward differentiating transcriptomic changes and understanding them was recently enabled by the development of accessible single-cell/nucleus sequencing techniques. Indeed, while earlier ALS/ALS-FTD CNS studies have documented robust transcriptomic changes that were often functionally associated with markers of the underlying cell types, recent studies have demonstrated successful denoising of the analysis by single-nucleus sequencing[17,18,20,21,29]. Here, we show that a hierarchical cell identity approach increases the power to detect differential genes and identifies genes that are changed in one or a few cell types, genes that are changed discordantly in different cell types, as well as genes which are changed similar in many cell types. Indeed, a large proportion of DEGs was regulated concordantly not only between cell types, but also across both disease groups, in line with a recent report[20]. This finding suggests that these changes are the result of a common factor acting on many cell types, e.g., genetics or a systemic environmental trigger. Our study included primary motor cortex samples both from donors with ALS and ALS-FTD, but not with FTD alone, likely explaining the discrepancy to a previous report of divergent transcriptomic changes

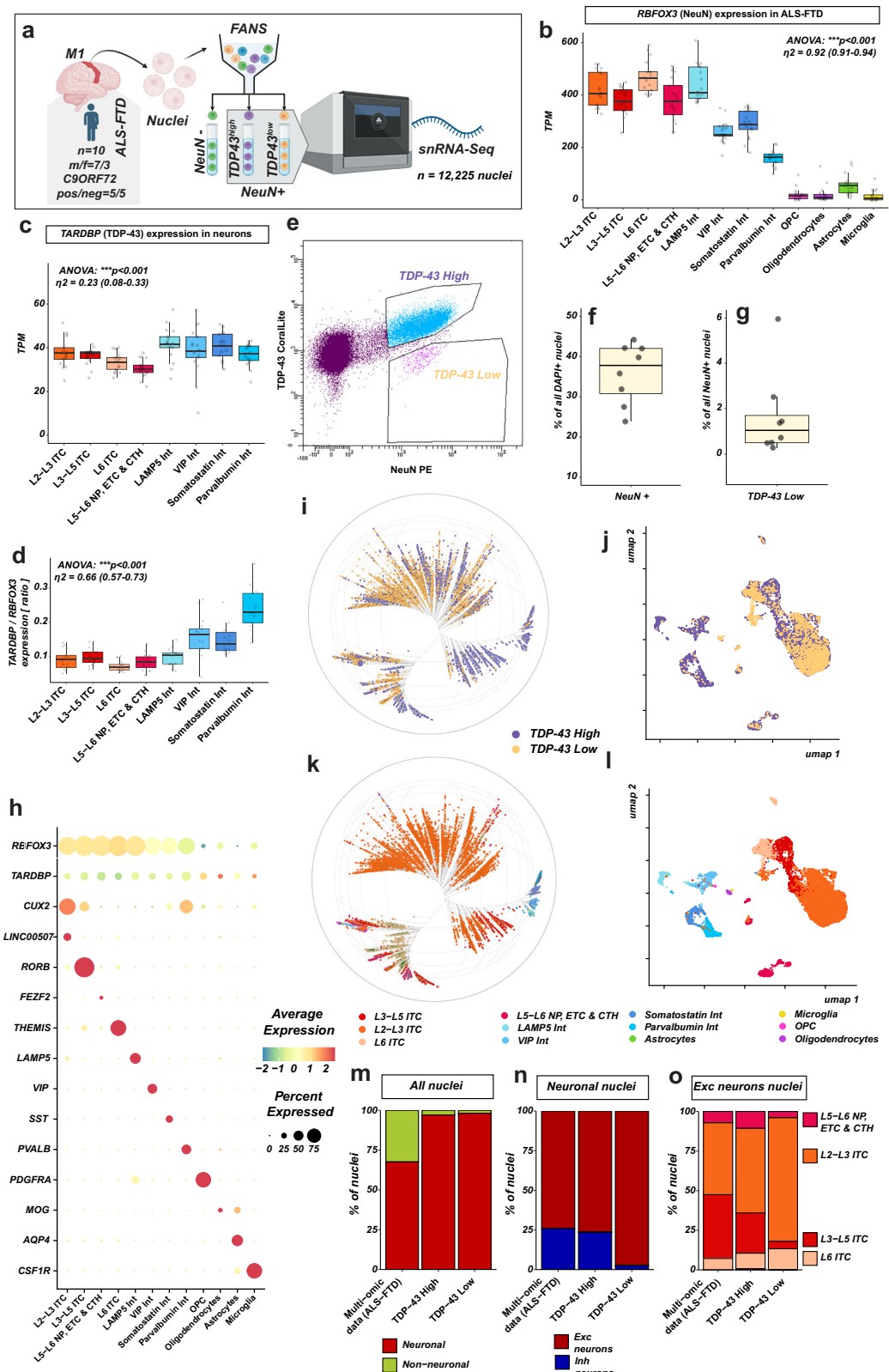

in the motor cortex between patients with C9-ALS and C9-FTD, but not including C9 ALS-FTD[21]. Consistent with Pineda et al.[20], our study revealed similar transcriptional changes between ALS and ALS-FTD, but stronger alterations in ALS-FTD based on both modalities transcriptome and chromatin accessibility. These differences were independent of *C9ORF72* HRE, as they were observed in both C9-fALS-FTD and sALS-FTD. In addition, oligodendrocytes were more affected in

ALS-FTD, aligning with histopathological findings of phosphorylated TDP-43 inclusions in oligodendrocytes in ALS-FTD. Using FANS-seq, we detected TDP-43 $^{Low}$ oligodendrocyte nuclei, despite initial NeuN + (*RBFOX3*) selection, supporting the extension of TDP-43 pathology beyond excitatory neurons. Such oligodendrocytes with expression of neuronal markers have been observed also previously[29]. The inclusion of these nuclei in the FANS-seq dataset facilitates the generalization of

**Fig. 6 | Fluorescence-activated nuclei sorting coupled with sequencing (FANS-seq) identifies cell types affected by TDP-43 pathology.** Flow-assisted nuclear sorting (FANS) followed by single-nucleus RNA-seq. NeuN + cell nuclei from 10 ALS-FTD motor cortex samples (m/f $n = 7/3$; fALS-FTD(C9ORF72)/sALS-FTD $n = 5/5$) were sorted into two fractions: TDP-43$^{Low}$ and TDP-43$^{High}$ (**a**). **b** *RBFOX3* (NeuN) gene expression in the ALS-FTD motor cortex (data from the multi-omic dataset, $n = 17$ samples, $n = 58{,}278$ nuclei). *RBFOX3* is expressed in all excitatory neurons and in LAMP5-interneurons (high expression) and in all other interneurons (lower expression), but not in glial cells, with very large overall effect size ($\eta^2 = 0.92$ (0.91–0.94)). **c**, **d** *TARDBP* (TDP-43) expression in different neuronal nuclei types (same dataset as in **b**) as TPM (**c**) and as a ratio to *RBFOX3* expression (**d**). Inter-neurons express higher levels of *TARDBP* mRNA than excitatory cortical neurons (overall effect size $\eta^2 = 0.23$ (0.08–0.33)). **b**–**d**: two-sided test statistics. **e** Gating strategy for the sorting of nuclei affected by nuclear depletion of TDP-43. Nuclei with medium and high NeuN+ signal (excitatory neurons and interneurons) were positively selected and sorted into two channels: proportional TDP-43 signal ("TDP-43$^{High}$", relative to the NeuN-signal) and strongly decreased TDP-43 signal (relative to the NeuN-signal, "TDP-43$^{Low}$"). **f** Distribution of % of DAPI + nuclei sorted into any channel in 8 flow cytometry runs. ~30–40 % of nuclei on average were positively selected. **g** Distribution of TDP-43$^{Low}$ nuclei as percentage of all neuronal nuclei in

the same 8 experiments as in (**f**). **h** Marker expression and nuclei clustering demonstrate the successful identification of cell types by label transfer from the high-resolution annotated multi-omic dataset. Bonsai representation (**i**, **k**) and UMAP (**j**, **l**) of cell distances and clustering by TDP-43 fraction (**i**, **j**) and cell type (**k**, **l**). Dot plot: percent of cells expressing the marker, average scaled, log-normalized expression. **m**, **n** FANS-seq sorting results in strong enrichment of neuronal nuclei. As expected, ~2/3 of all cortical nuclei are neuronal (multi-omic dataset), whereas FANS-seq sorted nuclei were 98% neuronal (TDP-43$^{High}$: 2.9 %, TDP-43$^{Low}$: 1.8 %). Of these, 23 % were from interneurons in TDP-43$^{High}$ (28 % in the multi-omic dataset) and 2.5 % in TDP-43$^{Low}$. **o** Selective enrichment of L2-3 intra-telencephalic and L6 intratelencephalic neurons in nuclei affected by TDP-43 pathology. "L5-6 NP, ETC & CTH": Layer 5 and 6 near-projecting, extratelencephalic and corticothalamic neurons. Effect sizes: eta squared ($\eta^2$) with two-sided 95% confidence interval and one-way ANOVA, two-sided ***$p < 0.001$. Box plots with median (center line), first and third quartiles (25/75%) and minimum/maximum value within 1.5 x interquartile range from the first/third quantile, respectively (lower/upper whisker). Source data are provided as a Source Data file. Icon art in a generated with BioRender.com (Created in BioRender. Lee, J. (2026) https://BioRender.com/e946rv8).

the TDP-43 pathology signature to cell types other than excitatory neurons.

Similarly to several previous single-nucleus studies, we did not observe a robust decrease in a specific *transcriptional* neuron type in this study. While Li et al. and Limone et al. also found no cell type changes, Gittings et al. found a reduction of L5 ETC neurons, but only in the C9-FTD frontal cortex, and not in the C9 ALS/ALS-FTD motor cortex. Similarly, Pineda et al. found a reduction of L5 ETC neurons only in the DLPF cortex with single-nucleus sequencing, but not in the motor cortex. In fact, the authors of this study conclude after comparing to IHC that sequencing is not appropriate for a precise cell composition analysis. Thus, our study is in accordance to previous reports.

### Limitations of the study

Our study comes with two main limitations related to two of the three main datasets that we utilized. For the multi-omic single-nucleus dataset, the major limitation is the number of cells that could be analyzed. With >180,000 nuclei that passed quality control, this dataset's size is comparable to or even larger than previous single-cell studies of the ALS/ALS-FTD motor cortex. However, the relatively low frequency of several cell types of interest precluded the possibility for an in-depth analysis of these cell types, especially in a study design like ours that aims to capture robust changes across a high number of donors. Examples include microglia, which were previously implicated in ALS/ALS-FTD through phenotypic changes and functional aberrations, and FEZF2-NTNG1 cells, which we identified as the Betz cells of layer Vb, and which are the canonical vulnerable cells in ALS/ALS-FTD (only ~500 cells in total detected). For the detailed study of these cells in the future, their numbers have to be enriched first, e.g., by FACS or MACS.

The second major limitation relates to the anatomical source of the spatial transcriptomics data. We utilized the LIBD spatial dataset of the DLPF cortex to map cell identities derived from a different area than the primary motor cortex. However, both areas are closely related histologically and implicated in ALS-FTD; furthermore, recent genomics studies including both areas in parallel have demonstrated a remarkable convergency of the molecular changes across the areas in ALS-FTD. For an optimal mapping of single-nucleus data to its histological correlates, future studies should generate spatial transcriptomic data from the human primary motor cortex, ideally including ALS-FTD patients to be able to identify cells with TDP-43 pathology by staining for phosphorylated TPD-43.

A further limitation of our study is that we did not consider different types of data that can be derived from the transcriptomic reads, e.g., differential splicing, nascent RNA and mature RNA. Such

analyses are emerging as important to refine the understanding of the transcriptome snapshot and should be employed in future analysis iterations of our data. Furthermore, expression and accessibility of transposable elements were not analyzed in this study, a factor that have been unequivocally associated to TDP-43 pathology. A further limitation is the sparsity of the ATAC-seq data, even though we profiled chromatin accessibility ~5–10-fold deeper than other studies. With a much larger feature set than RNA-seq (~200,000 peaks included in this study, around $1 \times 10^6$ peaks identified in total) and a higher percentage of reads 'lost' due to PCR duplication or mitochondrial reads, the costs of deep sequencing rise progressively for ATAC-seq. This hinders sophisticated analyses like genetic association with ATAC peaks or cQTL analysis, but will probably be alleviated in the future, with the continuously decreasing sequencing costs per read. Finally, it was not possible in this dataset to differentiate between ALS/ALS-FTD phenotypes (e.g., bulbar vs. spinal onset) or postmortem interval due to incomplete data. This is an important aspect which could be associated, e.g., with the lack of cell-type differences observed.

## Methods
### Human samples and ethics

All experiments with human material were performed in accordance with the Declaration of Helsinki, and the research approach has been approved by the Ethics Committee of the University of Ulm (*Nr. 135/20-FSt/TR*). Human flash-frozen postmortem tissues were obtained from: The Netherlands Brain Bank, Netherlands Institute for Neuroscience, Amsterdam (open access: www.brainbank.nl) (33 Samples), Edinburgh Brain and Tissue Bank (22 Samples), Massachusetts General Hospital (6 Samples), Laboratory for Neuropathology of Ulm University (8 samples) (Ethical approval No. 54/08), Hannover Medical School for Neurology (11 Samples), and Clinical Neuroanatomy of Ulm University (2 Samples). All material has been collected from donors for or from whom a written informed consent for a brain autopsy and the use of the material and clinical information for research purposes had been obtained. Clinical diagnosis was based on the standardized criteria set by each center (*further information available at each center upon enquiry*) and confirmed pathologically by the presence of pTDP-43 pathology, using immunohistochemistry as described in detail in Bolborea et al.[30] (Supplementary Fig. 1). Demographic, clinical and neuropathological information are provided in Supplementary Data 1 & 2. Complete demographic data (*ethnicity, PMI, onset/phenotype*) was not available for all samples. Biological sex was listed as reported from the cohort records, validated in silico from the sequencing data and referred to as 'sex'.

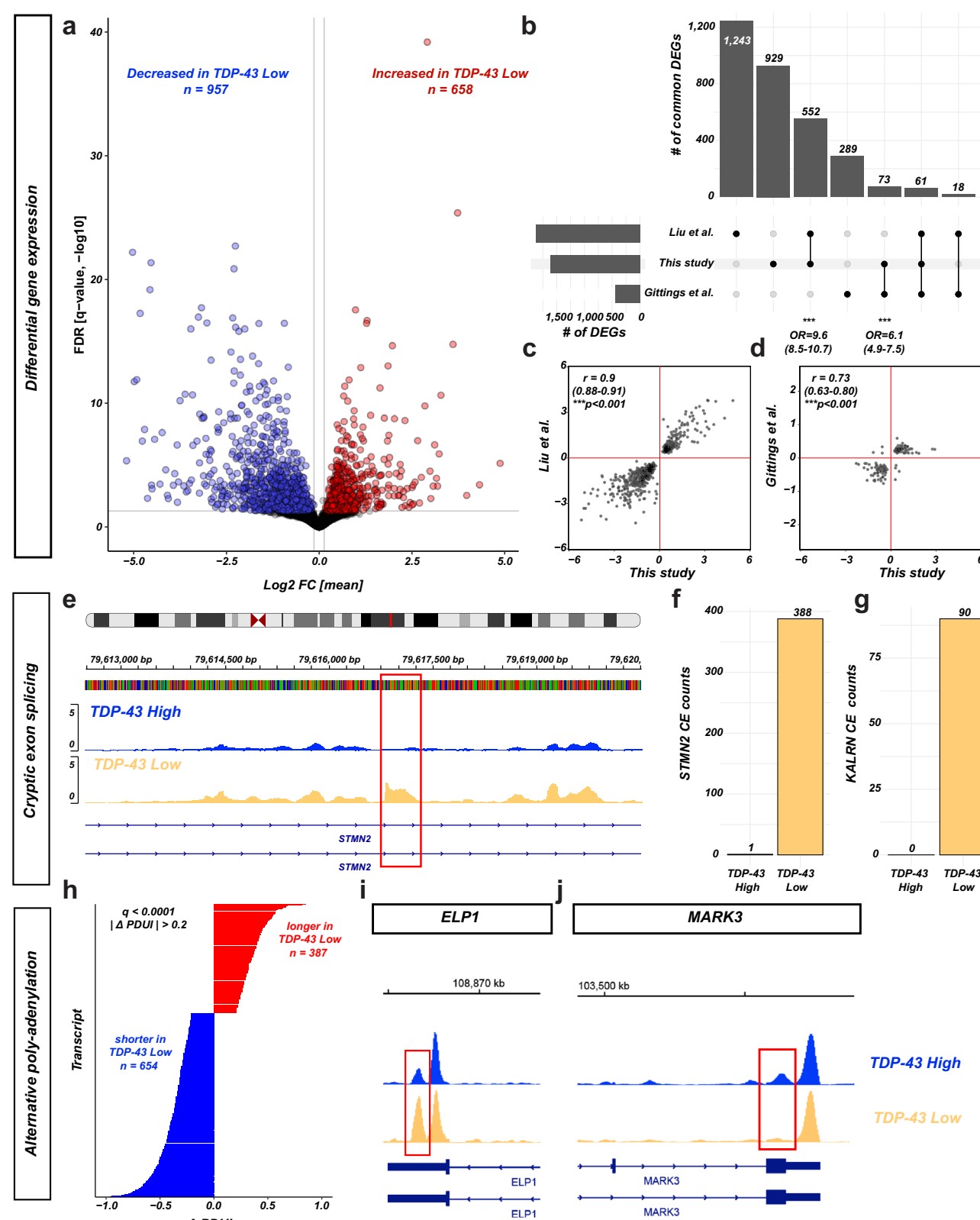

## Study design

The study includes 82 samples in two datasets: the multi-omic dataset and the FANS-Seq dataset. 72 samples are unique for the multi-omic dataset: ALS (n = 30), ALS-FTD (n = 10), and age- and sex-matched controls without neuropathology (n = 32). 7 samples were included in both datasets (ALS-FTD), and 3 samples are unique to the FANS-Seq dataset (ALS-FTD). The cohort/center information for the samples in the FANS-Seq dataset is not specified to preserve privacy (privacy concerns due to small sample size). The multi-omic dataset is comprised of 79 samples in the age range 33–96 years (mean: 66.53, median: 65.5, IQR: 56.25-77.75). One sample had missing age information. Age distribution was significantly different between the ALS and Control groups, and not significant between ALS-FTD and Control groups and between ALS and ALS-FTD groups (Wilcoxon rank-sum test with alpha 0.05). Male/female

**Fig. 7 | Transcriptional aberrations associated with TDP-43 pathology.**
**a** Differential gene expression in TDP-43 Low vs. TDP-43 High nuclei. Pseudo-bulk analysis with all nuclei per group irrespective of cell types. Volcano plot with fdr (0.05 threshold at gray line) and log2 fold changes. **b** Overlap of the differentially expressed genes (DEGs) identified in this study (DEGs from **a**) and the studies of Liu et al. (bulk RNA-seq of FANS-sorted TDP-43 Low vs TDP-43 High nuclei) and Gittings et al. (single-nucleus RNA-seq with imputed TDP-43 pathology by the presence of CEs). Upset plot with bars and numbers for each intersection. Odd ratios below both intersections of our study with each other study show the enrichment statistics for the overlap (OR with 95% CI, Fisher's exact test, ***$p < 0.001$).

**c** Correlation of gene expression changes for genes which were DEGs in our study and Liu et al. (**c**) or Gittings et al. (**d**). Pearson's ρ with 95% CI. ***$p < 0.0001$. Axes: log2fold changes. (**b–d**) two-sided test statistics. **e** Genomic track visualization of read coverage over the CE between exon 1 and 2 of *STMN2* (red box). Blue track: TDP-43 High nuclei, yellow track: TDP-43 Low nuclei. Coverage visualized with BigWig. **f, g** CE counts in each TDP-43 fraction in *STMN2* (**f**) and *KALRN* (**g**) genes. Numbers above bars = number of counts. **h** Alternative poly-adenylation events in TDP-43 Low nuclei. x-axis: Δ PDUI, y-axis: sorted transcripts. **i, j** Genomic tracks visualization of the alternative poly-adenylation events at 3′ UTR of *ELP1* and *MARK3*.

distribution in the whole cohort was 61%/39% and not significantly enriched in any case group (Fisher's Exact Test, $p$-value > 0.05). The FANS-Seq dataset is comprised of 10 samples. After sorting and sequencing, TDP-43-High nuclei were available from 10 of these samples and TDP-43 Low nuclei from 8 of these samples. The age range was 60–75 years (mean: 67.38, median: 66.50, IQR: 64.0-70.75). Age distribution was not significantly different between the TDP-43-High and TDP-43-Low groups (Wilcoxon rank-sum test with alpha = 0.05; groups ages are not identical because from two sample donors only TDP-43-High nuclei were successfully sorted and sequenced). Male/female distribution in the whole cohort was 70%/30% and not significantly enriched in one of the TDP-43 High/Low groups (70%/30% & 62.5%/37.5%, Fisher's Exact Test with alpha 0.05). Due to the heterogeneity of data collected at the different centers comprising the study cohort, more comprehensive population data could not be collected to be used for covariate modeling. Therefore, covariates were extensively modeled with surrogate variable analysis as indicated in the manuscript. Blinding was not possible during sample collection, as specific, matched samples had to be selected and pooled together. The investigators were blinded to group allocation during sample preparation, sequencing, and pre-processing of the data during data analysis. Blinding of the data analysts was not relevant for the downstream data analysis, as it is performed computationally, based on analysis statistics, and not by subjective judgment of the investigators.

RIN scores were used to evaluate bulk RNA quality for each sample before library preparation. All patients' samples were analyzed regarding pathogenic variants in 43 known ALS-associated genes (*ALS2, ANG, ARHGEF28, ATXN2, BSCL2, CCNF, CHCHD10, CHMP2B, DCTN1, ERBB4, FIG4, FUS, GBE1, GLE1, GRN, HNRNPA1, HNRNPA2B1, HSPB1, HSPB8, MAPT, MATR3, MME, NEFH, NEK1, OPTN, PFN1, PRPH, SETX, SIGMAR1, SOD1, SPG11, SPG20, SQSTM1, TAF15, TARDBP, TBK1, TUBA4A, UBQLN2, VAPB, VCP, VEGFA, VPS54*) using targeted gene sequencing. *C9ORF72* hexanucleotide repeat expansion (*HRE*) was investigated by repeat-primed PCR and Southern blotting. Samples were dissected from the primary motor cortex (*Brodmann area 4; BA4*).

## Nuclei isolation of human brain tissue
Nuclei were isolated from flash-frozen brain tissue, using an OptiPrep density gradient (Merck #D1556). For each donor, ~ 20 mg of tissue was dissected from the gray matter of the motor cortex using a dry-ice pre-chilled scalpel blade. Two to four tissue donors were pooled for the preparation of the single-nuclei suspension and loaded on a single well on the Chromium Next GEM Chip J (10X Genomics). Later, the individuals were demultiplexed based on genetic polymorphism (*see below*). The tissue (approximately 80 mg) was mechanically homogenized in a 7 ml glass tissue grinder (Merck, #D9063) on ice containing 1.4 ml of ice-cold homogenization buffer (320 mM Sucrose, 5 mM CaCl$_2$, 3 mM Mg(Ac)$_2$, 10 mM Tris-HCl (pH 8.0), 0.1 mM EDTA (pH 8.0), 0.1% NP-40, 1 mM β-mercaptoethanol, 0.4 μ/μl) using ~15 strokes with a loose pestle, followed by 15 strokes with a tight pestle, until all tissue parts were dissolved. The homogenized tissues were filtered through a 70 μm Flowmi cell strainer (Bel-Art, #136800070) and afterwards through a 40 μm strainer (Bel-Art, #136800040). Then, 700 μl of homogenate was mixed with 450 μl of

working solution (50% OptiPrep, 5 mM CaCl$_2$, 3 mM Mg(Ac)$_2$, 10 mM Tris-HCl (pH 8.0), 0.1 mM EDTA (pH 8.0), 1 mM β-mercaptoethanol) and layered on top of an OptiPrep density gradient consisting of 300 μl of 40% OptiPrep (40% OptiPrep, 96 mM Sucrose, 5 mM CaCl$_2$, 3 mM Mg(Ac)$_2$, 10 mM Tris-HCl (pH 8.0), 0.1 mM EDTA (pH 8.0), 0,03% NP-40, 0.12 U/μl RiboLock) at the bottom of a tube, followed by 750 μl of 30% OptiPrep (30% OptiPrep, 134 mM Sucrose, 5 mM CaCl$_2$, 3 mM Mg(Ac)$_2$, 10 mM Tris-HCl (pH 8.0), 0.1 mM EDTA (pH 8.0), 1 mM β-mercaptoethanol, 0,04% NP-40, 0.17 U/μl RiboLock). Nuclei were pelleted at the interface of the OptiPrep density gradient by centrifugation at 10,000 x $g$ for 5 min at 4 °C. The nuclear pellet was collected by aspirating ~ 200 μl from the interface and transferred to a 1.5 ml DNA-LoBind tube (Eppendorf, #0030108051). The nuclei were washed three times in 250 μl washing buffer (2% BSA containing 0.12 μ/μl RiboLock) by centrifugation at 2000 x $g$ for 3 min at 4 °C. During the final wash step, the nuclei suspension was again filtered through a 40 μm Flowmi cell strainer to remove debris and aggregates. After the last centrifugation step, the pellet was re-suspended in 50 μl of 1X nuclei buffer (1X nuclei buffer (20X stock provided from 10X Genomics, #2000207), 1 mM DTT, 1 μ/μl Ribo-Lock). As a quality control, the number and quality of the nuclei was assessed upon DAPI staining in a hemocytometer.

## Generation of multi-omic single-nuclei libraries and sequencing
Multi-omic single-nuclei ATAC and gene expression libraries from the same nuclei were prepared using the Chromium Next GEM Single Cell Multiome ATAC + Gene Expression reagents (10X Genomics, #PN-1000283). Sample processing and library preparation was performed according to manufacturer instructions (user guide CG000338 RevF, from 10X Genomics). Two to four samples were multiplexed per single 10X Chip well (Supplementary Data 3). Single-nuclei ATAC libraries were sequenced using the NovaSeq 6000 SP PE150 (Illumina), and single-nuclei gene expression libraries were sequenced on a NovaSeq 6000 S4 PE150 (Illumina).

## Isolation of genomic DNA for genotyping and demultiplexing
A piece of tissue of approximately 15 mg was dissected with a scalpel blade and supplemented with 500 μl nuclear lysis buffer (*2.5 M NaCl, 0.5 M EDTA*), 10 μl proteinase K (*10 mg/ml*) and 20 μl of 20% SDS. The sample was incubated overnight in a gently shaken water bath. The next day, 180 μl of 6 M NaCl solution was added, and the tube was shaken vigorously. The sample was centrifuged twice at 1500 x $g$ for 15 min at room temperature. After each centrifugation, the supernatant was transferred to a new tube. Then, 2 volumes of absolute ethanol were added to the supernatant. Careful inversion of the tube allowed the DNA to condense and to become visible. The DNA strand was washed briefly in 70% ethanol and transferred to a tube containing 30 μl of water. The DNA was incubated overnight at 4 °C until it was soluble enough for quantification on the NanoDrop One.

## Genotype-based sample demultiplexing
To reduce batch effects and facilitate doublet removal by genotype identification, we pooled two to four sample donors for each 10X Multiome reaction well. Later, we identified the donor of origin of each

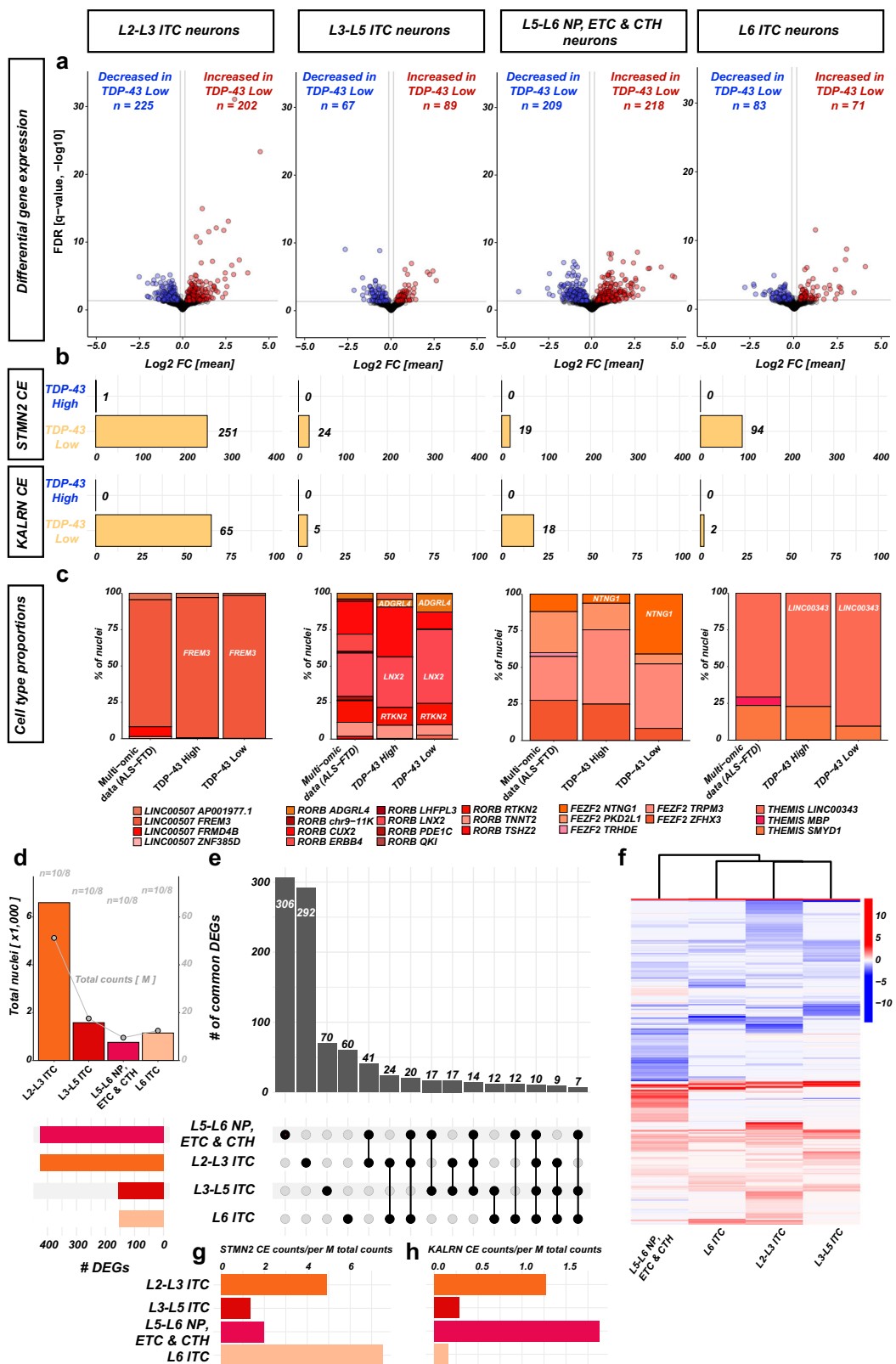

**Fig. 8 | Cell-type specific effects of TDP-43 pathology.** Analysis of differential gene expression and CE inclusion in *STMN2* and *KALRN* in the four major excitatory cell types. **a** Volcano plots with log2 fold-changes (x-axis) and fdr (y-axis, line at 0.05). **b** Read counts for junctions over CE in *STMN2* and *KALRN* in each major cell type. **c** Proportions of cell types in each major cell type in the whole multi-omic dataset as reference (left bars), the TDP-43 High fraction (middle bars) and the TDP-43 Low fraction (right bars). Frequencies as % of all parent major cell type nuclei. **d** Cell/read coverage for each major cell type in the FANS-seq dataset. Total nuclei

(bars & left y-axis, x1000) and total read counts (gray points with line, right y-axis, in 10^6 UMIs). **e** Upset plot demonstrates that the majority of DEGs are specific to one major cell type. **f** Heatmap of all DEGs from (**e**) with scled log2 fold-changes (color scale). Hierarchical clustering with average linkage and Euclidean distance. **g**, **h** CE counts in *STMN2* and *KALRN* as a ratio of CE counts to total counts (in 10^6) for each major cell type demonstrate the specificity of CE inclusion.

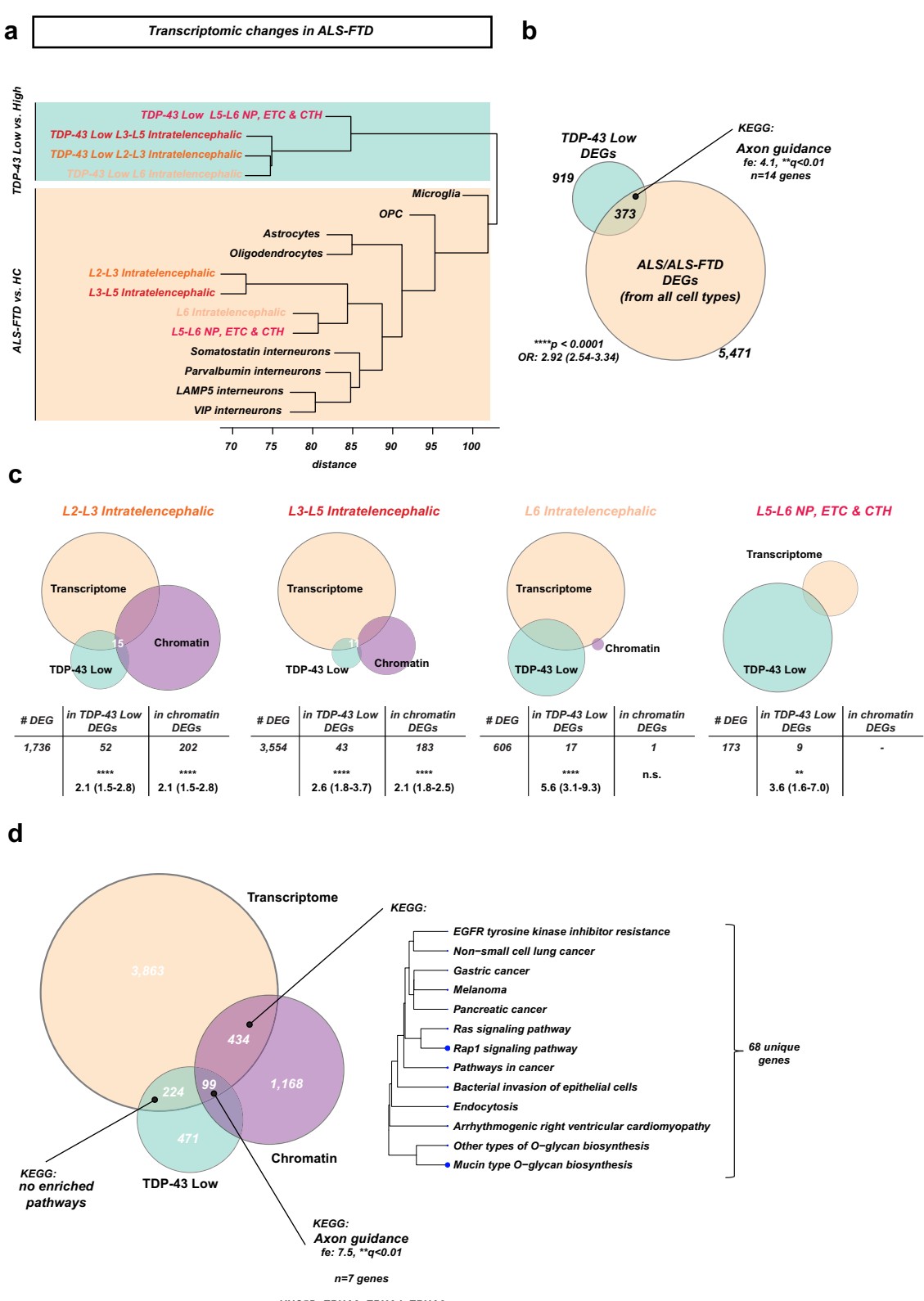

nucleus in the dataset with a combination of in silico and in vitro sex- and SNP-based demultiplexing. For sample demultiplexing, we used Souporcell, a genotype-free bioinformatics tool to perform SNP-based demultiplexing[31]. First, reads are remapped using minimap2 and variants are called using freebayes. Vartrix is then used to count alleles per cell. Finally, cells are clustered based on the number of reads

supporting each allele of the called variants. Cluster (i.e., donor) SNPs are then analyzed and SNPs and SNP combinations that uniquely identify a donor were genotyped on DNA isolated from each sample separately by Sanger sequencing after PCR with primers flanking the SNPs from both sides. DNA was purified from the same motor cortex samples that were used for the generation of the single-cell data.

**Fig. 9 | Association of transcriptomic changes in the ALS/ALS-FTD motor cortex with TDP-43 pathology and epigenetic changes. a** Hierarchical clustering of transcriptomic changes in the ALS-FTD motor cortex and in TDP-43 $^{High}$ vs TDP-43 $^{Low}$ nuclei (also from the ALS-FTD motor cortex) demonstrates that these are two distinct sets of transcriptomic changes. Cell type comparisons were clustered by the scaled log2 fold-changes for the set of all genes which were found significant in at least one of these comparisons. Hierarchical clustering with Euclidean distance. Excitatory neuron major cell types highlighted. **b** Overlap of the TDP-43 $^{Low}$ DEG

signature (i.e., all DEGs from the four major cell types with the ALS/ALS-FTD DEG signature (i.e., all DEGs from at least one major cell type in at least one disease group). Overlap is significantly enriched (***$p$ < 0.0001, Fisher's exact test with OR and 95% CI, two-sided). **c** Overlap of DEGs in motor cortex, TDP-43 $^{Low}$ vs TDP-43 $^{High}$, and differential accessibility analysis for each major cell type in the ALS-FTD cortex. **d** Same overlap for all DEGs from **c** together. Enriched KEGG terms (fdr < 0.05) are listed for the overlaps of differentially accessible chromatin-associated DEGs and TDP-43 pathology-associated DEGs.

## Fluorescence-activated nuclei sorting (FANS)

Nuclei isolation and fluorescence-activated sorting were performed based on a protocol by Policicchio et al. (*dx.doi.org/10.17504/protocols.io.bmh2k38e*) and Liu et al.[10] with adaptations. Approximately 300 mg of primary motor cortex tissue was homogenized in 3.2 ml of homogenization buffer, and the total homogenate, together with the working solution, was distributed onto four OptiPrep gradients. Nuclei from the different OptiPrep gradients were pooled during the transfer from the interface to a new tube. Following the washing steps, the nuclei were finally resuspended in 1 ml staining buffer (*PBS supplemented with 0.5% BSA and 0.12 µ/µl RNase inhibitor*) and incubated for 15 min on ice. Nuclei were then stained with fluorescently-labeled antibodies targeting TDP-43 (*1:500, CoraLite488-conjugated TDP-43, #CL488-10782, Proteintech*) and NeuN (*1:500, Milli-Mark™ Anti-NeuN-PE antibody, #FCMAB317PE, Sigma-Aldrich*). Further, nuclei were stained with 1 µg/ml DAPI (*#D9542, Sigma-Aldrich*) to identify intact single nuclei. In addition, 5% of the total nuclei were stained with one of the respective antibodies or DAPI to be used as single-stained controls in subsequent flow cytometry experiments. Staining was performed in staining buffer for 90 min at 4 °C in a rotating wheel. After incubation, nuclei were pelleted by centrifugation at 2000 $x\,g$ for 3 min at 4 °C, re-suspended in 500 µl staining buffer and subjected to FANS performed by the Cytometry Core Facility at the University of Ulm using a FACS Aria II SORP (*BD*). Single neuronal nuclei (*NeuN +*) were sorted for either TDP-43 high or low content. Immediately after sorting, single-nuclei gene expression libraries were prepared according to the Chromium Next GEM Single-Cell 3' Reagent kit v3.1 user manual (*CG000315 RevF, from 10X Genomics*). The libraries were sequenced on a *NovaSeq X 25B with PE150*.

## FANS RNA-seq data pre-processing

FANS-seq data was demultiplexed and converted from BCL files to FASTQ files with bcl2fast2 (*Illumina*). FASTQ files were processed with Cellranger count (v9.0.1) (10X Genomics) with the GRCh38-2024-A genomic reference (*10X Genomics*), including introns. For each sample, ambient RNA signal was checked with SoupX[32], corrected with CellBender (*v0.3.0*)[33] and converted to Seurat-compatible h5 files with PyTables (*v3.10.2*). Nuclei doublets were detected with scDblFinder (*v1.20.2*) with a number of artificial doublets, which provided stable doublet detection for all samples[34].

## Multi-omic sequencing data pre-processing

BCL files were demultiplexed and converted to FASTQ files with bcl2fast2 (Illumina). FASTQ files from both single-nucleus ATAC-seq (*snATAC-seq*) and single-nucleus RNA-seq (*snRNA-seq*) were processed using the 'count' function in Cellranger-arc (*2.0.2*) using the reference GRCh38-2020-A-2.0.0 (*10x Genomics*). RNA count matrices were merged using Seurat (*v.4.4.0*)[35]. To identify accessible chromatin regions, peak calling was performed using MACS2 (*v.2.2.6*)[36] with the options '--extsize 200, --shift -extsize/2'. Consensus genomic regions ('*peaks*') were generated with Signac (*v1.10.0*)[37] with the 'reduce' function and filtered to retain only genomic regions found in at least 5 samples. Peaks < 30 base-pairs (*bp*) or > 2500 bp were excluded.

## Nuclei filtering

Multi-omic nuclei were filtered in the merged dataset according to the following parameters per nucleus: Total number of RNA counts

> 1000, total detected fragments > 1000, ratio of mono-nucleosomal to nucleosome-free fragments < 3, transcription start site (*TSS*) enrichment score > 1, fraction of ATAC-seq reads in peaks > 0.2, percent of GEX-reads in mitochondrial genes < 10%, percent hemoglobin counts < 1 % and percent of reads in ribosomal genes < 10% (Supplementary Fig. 2). Filtering of the FANS-seq dataset was performed in the merged dataset, with median absolute deviation (*MAD*) filtering: > 5 MAD of log1p RNA counts, percent of reads in the mitochondrial genes in the uncorrected data < 5% and percent of reads in ribosomal genes < 5%.

## Data integration

For data integration and batch correction of the multi-omic dataset, we utilized the single-cell variational inference (*scVI*) toolkit[38,39]. scVI uses probabilistic modeling and a latent variable framework to correct for batch variation in single-cell RNA sequencing data, minimizing technical artifacts across the experimental batches. We included the percentage of mitochondrial reads as a continuous covariate and sex as a categorical covariate, as gonosomal genes were retained. To tune the hyperparameters, we utilized a Gaussian process-based optimization with the 'gp_minimize' function from scikit-optimize[40]. We made two adjustments to the default settings, resulting in an optimized Evidence Lower Bound (*ELBO*) score. Specifically, we increased the number of latents to 50 and the number of nodes per hidden layer to 200. For the ATAC model, we employed PeakVI, a deep generative model for single-cell chromatin accessibility analysis[41]. The model was set up with 'atac_peak_region_fragments' as a continuous covariate. The best ELBO score was obtained increasing the number of latents to 50 and the number of nodes per hidden layer to 332. The covariates were included in all layers of the decoder. All other parameters were kept at their default settings. The FANS-seq dataset was integrated with Harmony[42] and the SCT-transformed data with Seurat implementation's default parameters[35].

## Cell clustering, dimensionality reduction and cell-type annotation

Dimensionality reductions were computed with PCA, UMAP[43] (Supplementary Fig. 3) and Bonsai-Scout[23]. For the multi-omic single-nucleus dataset, UMAP was calculated on the basis of the 50 latent variables from the scVI model (*RNA-seq*), the peakVI model (*ATAC-seq*), or both (*WNN analysis*); for the FANS-seq dataset, on the first 30 PCA components of the normalized data. For Bonsai-Scout, the clustering was performed on the whole transcriptomic count matrix and randomly sampled 10,000 nuclei for the multi-omic dataset, and on the whole transcriptomic count matrix for the FANS-seq dataset. Cell types were annotated manually by generating lists of differential cell type markers and curating cell types based on the current literature, foremost comparing to Bakken et al.[19], Pineda et al.[20] and Gittings et al.[18]. Cell-type gene expression and chromatin accessibility markers were calculated for each hierarchical cell type against all other cells using only nuclei from the neurologically unaffected group (*Ctrl*) with Seurat (*'FindMarkers' function*) and presto (https://github.com/immunogenomics/presto) with Seurat's Wilcox test implementation[35]. Hierarchical cell-type annotation to major cell types and cell type classes was performed manually (Supplementary Fig. 4). Data from unsupervised clustering is provided in Supplementary

Figs. 5–7. For visualization of marker feature expression in the major cell types (Fig. 1f), metacells were constructed with Cicero using the WNN UMAP reduced coordinates and 100 k-nearest neighbors[44]. 100 metacells were then randomly sampled per group and plotted. For visualization of the chromatin accessibility profiles across the major cell types (Fig. 1g), the cumulative, scaled accessibility was plotted for each cluster with Signac (*CoveragePlot*). For the hierarchical clustering of the major cell types based on the 'average' cell per *group* (Fig. 1f, g), all cells and all features (*genes or genomic regions*) were used, and the dendrogram plotted with Seurat (*BuildClusterTree/PlotClusterTree*) with the default metric and method (*Euclidean distance, complete linkage*). FANS-seq dataset was annotated by label transfer from the multi-omic dataset with Seurat with 'FindTransferAnchors' and 'TransferData' functions with log1p-normalized data, PCA, 30 principle components and 'k.anchor = 5'. Discordant hierarchical annotations of the same nucleus were preserved for downstream analysis for the respective hierarchical annotation levels.

## Compositional data analysis

The compositional analysis to identify changes in the relative cell-type proportions across the disease groups was performed with scCODA, a Bayesian-model-based classifier which outputs a measure of the likelihood that a specific cell type or a node in the hierarchical cell-type tree is specifically changed as opposed to passively due to the compositional nature of the data[45]. We simulated compositional changes with automatic cell type reference choice and the default parameters. Furthermore, we analyzed compositional cell-type data with mixed linear modeling based on MASC[46].

## Differential expression/accessibility testing

To capture the transcriptomic perturbations associated with ALS/ALS-FTD across different cell types, we utilized two complementary approaches for modeling of differential gene expression in combination with the hierarchical cell type annotation: 'pseudo-bulk' and model-based single-cell analysis. The pseudo-bulk analysis is based on the aggregation of all counts from a given cell type per sample (*person*) and captures robustly transcriptomic changes that are characteristic for the disease across different patients; the model-based approach benefits from the increased power of single nuclei as observations and captures transcriptomic changes with increased sensitivity, but reduced power to generalize across patients. The combination of both methods allows for sufficient sensitivity at high resolution (*i.e., comparison of small clusters/rare cells*) and improved specificity at low resolution (*larger cell clusters, like major cell types*), while decreasing the disadvantages associated with both approaches. We used surrogate variable analysis (*SVA*)[47] to compensate for known and hidden batch effects (*e.g., PMI, onset site, cohort*) in pseudo-bulk, and used fixed plus random effects to accommodate sample heterogeneity and batch effects in the mixed models. We further used sample label permutations and random gene samples as negative controls to control the specificity of the results.

Differential expression/accessibility of pseudo-bulk samples was performed with DESeq2[48] after sample-wise (*i.e., donor*) aggregation of the raw counts for the respective cell type analyzed. Then, differential expression/accessibility was performed with Wald's test with covariates as specified in the results section. We estimated the gene-wise dispersion in a multigroup fashion (*Ctrl + ALS + ALS-FTD instead of Ctrl + ALS and Ctrl + ALS-FTD separately*) to increase the stringency and specificity of the analysis: DESeq2 analysis was run on all samples together and then the results extracted for each disease group vs. controls comparison. For SVA, surrogate variables (*SVs*) were extracted with the sva package and the SVs included as covariates in the DESeq model. For ranking, visualization, and correlation analyses, log2-fold changes were shrunken with apeglm[49]. Hierarchical DGE analysis was performed with the BulkR package (https://github.com/vgrozd/BulkR). Subsetting of FASTQ and BAM files for control analyses

was performed with seqtk (https://github.com/lh3/seqtk) and 10X Genomics' 'subset-bam_linux' (https://github.com/10XGenomics/subset-bam), respectively.

## Functional analysis

The annotation of transcripts to gene types was acquired from the ENSEMBL database accessed over the Bioconductor ecosystem (*Ens.db.Hs.86' and 'org.Hs.eg.db'* packages). Importantly, the genes which are expressed in the human motor cortex are heavily annotated to pathways and KEGG/GO terms related to neurodegenerative diseases (Supplementary Fig. 10). All enrichment analyses were therefore conducted against a background (*"universe"*) of the respective cell type/dataset. Comparison of sets of differential expression results were performed with the 'BulkR' R package (https://github.com/vgrozd/BulkR). Enrichment analyses were performed with ShinyGO[50], StringDB[51], fgsea[52] and clusterProfiler[53] R packages. Gene type annotations and alternative names between the 10X CellRanger reference genome versions 2020-A and 2024-A were converted with the specification files provided with the 10X CellRanger reference genomes' release notes.

Gene/genomic region module expression resp. accessibility scores were calculated with Seurat (*"AddModuleScore"*) against a background control with random features (*n = 500 unless specified otherwise*) with the default random seed (*1*). For cell-type specific module scores, only markers with q-value (*fdr*) < 0.05 and fold change > 5 (*up-regulated*) were used, and only markers for which both HGNC gene symbol and an ENSEMBL gene id were available were kept to allow for the simultaneous use of the markers for the single-nucleus multiome dataset and the human cortex spatial transcriptomics dataset. Data was handled with tidyverse and Pandas, GenomicRanges, magrittr, qs/qs2; single-cell datasets with Seurat, Scanpy and SingleCellExperiment. Data was visualized with the R packages ggplot2, complexHeatmap, pheatmap, ggrastr, complexUpset, ggnewscale, patchwork, scattermore, ggraph, data.tree, igraph, networkD3, ggrepel, ggpubr, dendextend, and with the IGV browser.

## Cryptic exon and alternative poly-adenylation analysis

Cryptic splicing analysis was performed as described in Gittings et al.[18]. Exon junctions were extracted from BAM files from CellRanger 'count' output with regtools' 'junction extract' with a minimum anchor length of 6 bp, minimum intron size of 30 bp and a maximum intron size of 500,000 bp[54]. Introns were then clustered with Leafcutter's 'leafcutter_cluster_regtools.py = m10 -p0.0001'[55] and counts for the CE-specific junctions counted. For cell-type or group analyses, barcoded BAM files were split with sinto (https://github.com/timoast/sinto) 'filterbarcodes' and merged with samtools' 'merge'[56]. Bedgraph files for visualization were generated with bedtools' 'genomecov'[57]. Alternative poly-adenylation analysis was performed with DaPars' 'python DaPars_main.py'[58] with the default parameters.

## Analysis of publicly available transcriptomics data

For spatial transcriptomics, we accessed the publicly available LIBD dataset[22] over the accompanying spatialLIBD package[59]. The dataset consists of 12 slides processed with 10X Visium Spatial from 3 neurologically unaffected controls, 2 pairs 300 μm apart from each other, and has been extensively characterized before[22,59]. Histological layers L1-L6 & white matter (*WM*) have been annotated by Maynard et al.[22]. Spatial gene expression and score heatmaps were plotted with the spatialLIBD[59] and ggplot2 packages.

For comparison of the differential gene expression (*DGE*) analysis results between the study of Liu et al. and ours, we performed the DGE analysis since the list of original results could not be accessed. In brief, we queried the dataset with the 'GEOquery' R package[60], aligned the reads with ENCODE's standard parameters for RNA-seq with STAR[61], and generated a count matrix with R Rsubread's 'featureCounts'[62]. DGE was then performed with DESeq2 with sex as a covariate.

## Annotation and linkage of chromatin accessibility

Chromatin accessibility was correlated to gene expression across all nuclei in the dataset with Signac (*'LinkPeaks' function*) with the default settings and a 500 K bp window from both ends of each peak. Chromatin peaks were annotated as a genomic element with the ChIPseeker package[63] (*'peakAnno' function*) and summarized to one of 6 categories: *'Intergenic', '5'UTR', 'Exon', 'Intron', '3'UTR' and 'Promoter'*, the last comprising peaks up to 3 K bp around the TSS.

## Statistics and reproducibility

Sample sizes were estimated from previous studies and recommendations for genomics studies. No statistical method was used to predetermine sample size. No data were excluded from the analyses. Vascular and immune cells were excluded due to low numbers that prevent proper analysis. The experiments were randomized in two ways: a random label permutation control was used were applicable as a 'negative' control; multiple samples were pooled per reaction well and random anonymized IDs assigned to samples after demultiplexing. The investigators were not blinded to allocation during experiments and outcome assessment. To increase reproducibility, multiple samples from different centers were included in the study cohort and raw data, processed data and data analysis code are provided with the study. Random seeds used for sampling and centroid/clustering initiation are specified in the source code and methods. For methods for which the random seed is not specified, the default random seed implemented in these methods/algorithms was used. We used sample permutation as a 'negative' control for unspecific results by supervised mixing of sample labels, i.e., random permutations were generated with a constraint such that the number of disease cases and controls are balanced in each random group and that the proportions of sex labels is comparable to the case-control comparison. Statistical overrepresentation was tested with Fisher's exact test. For all analyses with multiple target variables, the false discovery rate was controlled with the Benjamini-Hochberg adjustment of p-values[64]. Findings were considered 'statistically significant' with an fdr $q$-value < 0.05 until stated otherwise. ANOVA was performed with the R 'stats' package, effect sizes with the R packages 'effsize' and 'effectsize' and confusion matrices with accuracy with the R 'caret' package.

## Reporting summary

Further information on research design is available in the Nature Portfolio Reporting Summary linked to this article.

## Data availability

The raw sequencing data that was generated in this study is available on EGA under EGAD50000002240 (https://ega-archive.org/studies/EGAS50000001562) (multi-omic dataset) and EGAD50000002243 (FANS-seq dataset) and is subject to controlled access due to data privacy regulations (identifiable genomic sequence data in combination with phenotype). Requests for accessing the raw data are to be addressed to the Data Access Committee (EGAC50000000856) (https://ega-archive.org/dacs/EGAC50000000856) through the EGA platform. Requests will be reviewed by the Data Access Committee (expected timeframe: 25 working days), approved for use (in five years timeframe) if the application requires access to the raw genomic sequences and requires the signing of a data transfer agreement to guarantee compliance with the European General Data Protection Regulation (GDPR). Further requests for custom analysis without access to the raw data (i.e., collaboration) can be made to the corresponding authors per e-mail. The processed single-nucleus sequencing data with associated metadata is available on Zenodo under 10.5281/zenodo.18370645 (https://zenodo.org/records/18370646) (multi-omic dataset) and 10.5281/zenodo.18371192 (https://zenodo.org/records/18371193) (FANS-seq dataset). The publicly available data was accessed from the provided resources in Maynard et al.[22,59], Pineda et al.[20], Wang et al.[17], Li et al.[21], Limone et al.[29], Liu et al.[10] and Gittings et al[18]. Source data are provided in this paper.

## Code availability

The code generated for the analysis and the code used to generate the figures is available on GitHub: https://github.com/DanzerLab/Ruf_et_al_2026.

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

## Acknowledgements
This work was supported by the Deutsche Forschungsgemeinschaft (DFG) Emmy Noether Research Group DA 1657/2-1, SFB1506 (Aging at interfaces), SFB1149 (Danger Response, Disturbance Factors and Regenerative Potential after Acute Trauma) and Target ALS grant No. BM-2024-C2-L4. The authors acknowledge support by the state of Baden-Württemberg through bwHPC. We thank the authors of the LIBD spatial data and the associated tissue donors for the publicly available data. We thank the Netherlands Brain Bank (Amsterdam, the Netherlands) and the MRC UK Brain Bank for providing brain tissue from patients and controls. We would like to thank the Core Facility Cytometry of the Medical Faculty at Ulm University for providing support and instrumentation, BD FACSAria III funded by the Deutsche Forschungsgemeinschaft (DFG, German Research Foundation), project number 162388165. We acknowledge the expertise and assistance of Kathrin Müller, Joao de Meirelles and Ramona Bück. The authors thank the patients and their families for their brain donations. We thank the Edinburgh Brain Bank (EBB) for receiving brain material from patients and controls, including available clinical data. The EBB is supported by the Medical Research Council. The Edinburgh Brain Bank is a Medical Research Council funded facility with research ethics committee (REC) approval (11/ES/0022). DRT received support from the KU-Leuven internal funding Grant No. C14/22/132 and from Target ALS Grant No. BM-2024-C3-L4.

## Author contributions
W.P.R., J.K.K., S.J.B., L.M., and V.G. performed the experiments; W.P.R., J.L.B., and V.G. performed the computational biology analysis; G.S.V., D.Y.H., S.P., N.B.B., D.R.T., and K.M.D. generated the study cohort; D.Y.H., S.P., and D.R.T. performed the pathological characterization of the samples; W.P.R., J.K.K., V.G., and K.M.D. conceptualized the study and interpreted the data; W.P.R., J.K.K., V.G., and K.M.D. wrote the manuscript.

## Funding

## Competing interests
D.R.T. received a consultant from Muna Therapeutics (Belgium), and collaborated with Novartis Pharma AG (Switzerland), and GE-Healthcare (UK). The remaining authors declare no competing interests.
