## [Transparent Peer Review file · Nature Communications]

Multi-modal dissection of cell-type specific TDP-43 pathology in the motor cortex

Corresponding Author: Professor Karin Danzer

Version 0:

Reviewer comments:

Reviewer #1

(Remarks to the Author)

The authors have done a very thorough job in generating and describing single-nucleus multi-omic data from the motor cortex of ALS and ALS-FTD patients, alongside FANS sorted single-nucleus RNA-seq to identify nuclei with low TDP-43. Both these datasets will be tremendously useful to the wider field. All analyses have been done carefully and using the current state of the art tooling. I have only a few suggestions.

Given that the authors generated genotyping data to aid with demultiplexing of the single-nucleus data, it was a shame that this data wasn't incorporated into the paper, for example by looking for ALS-associated common variants that overlapped their chromatin accessibility regions, or even mapping QTLs if the sample size allows. Is this something the authors would consider?

The enrichment of neurodegenerative disease KEGG pathways in the DEGs used throughout the manuscript is likely due to all of those "disease" pathways having strong overlaps with the genes in the Oxidative Phosphorylation KEGG pathway. I would suggest the authors either remove these assertions, or refine it by testing only the genes that are unique to each disease pathway. Alternatively, the authors could perform enrichment with lists of GWAS and rare variant-linked genes for ALS.

Were the authors able to assess cryptic splicing of STMN2 and KALRN in the TDP-43 low fraction of single cells, given that those two genes have been previously demonstrated to be detectable in single cell data? PMID: 37466726.

Similarly, did the authors investigate whether NPTX2 was upregulated in the TDP-43 low fraction, given its recent demonstration as a TDP-43 target? PMID: 38355792

Minor comments:

Please use commas as thousand separators throughout.

Line 42 - STMN2 typo

Line 56 - the vast majority of polymorphisms detected by GWAS are in non-coding regions, I would change "some" to "most".

Fig. 4 - typo, ALS-FTD has 17 donors, not 30

(Remarks on code availability)

Code appears well organised. No README provided which would be helpful.

Reviewer #2

(Remarks to the Author)

This manuscript describes concurrent RNA and chromatin accessibility sequencing of single nuclei from ALS, ALS-FTD and

control motor cortex. This dataset emerges as a few other single cell datasets have been published, although this is the first using the technology that allows for both transcriptome and epigenome profiling in the same cell. The analysis pipeline is sophisticated, and a large number of cases are studies. However, the results are sometimes inconsistent with what is known which may be driven by relatively low coverage and/or cells per case. Moreover, while this is an impressive amount of data and analysis, I am not sure what additional insights these analyses have put forward towards our understanding of ALS and ALS/FTD. These issues include the following:

1. The cases/demographics file is not well formatted and is incomplete. In addition to demographic information, the reader needs information on the number of cells, number of reads per cell, number of cell types, etc. on a per case basis. While it is impressive that there is a large number of cases that were included, it does not seem that each case was sequenced to equal depth when examining figure 2d. Based on the number of cases and the number of total nuclei, the average number of cells per case is on the order of 2000-2500, with many cases likely being much lower, as evidenced by figure 4a where some cases have no nuclei from some cellular subtypes. This is problematic because some of the analyses are likely underpowered if there are not sufficient numbers of cells per case (and the number of cells per case is used as a variable in some analyses).
2. The lack of differences in proportion of various cell types in disease versus control is worrisome that this dataset is not capturing the molecular events in disease. This may be related to the low number of cells per case (point #1).
3. Figure 2A contains the data shown in figure 1D but broken up into different parts of the figure. As this data is repeated and is not really contributing additional data to figure 2, and since the breaking up of the UMAP further distorts the UMAP which is already non-linear, I suggest not including these dot plots.
4. Page 18 lines 438-439 states that the data was compared to "previous studies" which I do not see in the supplementary figure 11 – unless they mean just mapping their data using GSEA to KEGG pathways. This should be clarified.
5. As two large single cell datasets have been published in the last couple of years on this topic, a comparison of findings across datasets would be important to show either consistent (or discordant) results.
6. Figure 3 is problematic and appears to show that this analysis pipeline, matching single cell transcriptomes to existing spatial transcriptomic data, is not working properly. RORB is known to be a layer 4 gene, but they report it highest in layer 5. Similar problems exist for Themis (layer 5, not 6), FEZF2 (layer 5, not 6), SST (layers 2,3,5, not layer 1), microglia (equally dispersed throughout the CNS). This is important because the same methods are used later to map the authors' new data – and so if the method cannot do this accurately for some known genes (such as Figure 4N), it casts doubt on the new data localization.
7. Figure 3f shows a single dot in a very small inset as evidence of localization of Betz cells. This is not convincing.
8. The Venn Diagram in figure 4N, while significantly "enriched" in terms of overlap (likely due to the high statistical power when compared against the entire transcriptome), is striking for how the vast majority of the results do not overlap.
9. Page 21, line 508, states that DARs "correlated strongly". While some p-values are very low due to the high number of datapoints, the r values (which is used to judge the strength of a correlation) are very low suggestive of a very weak correlation.
10. Figure 5i, can the authors clarify on which dataset the Spearman correlations were performed? It appears to be just for the data that is both differentially expressed and differentially accessible which is fine but just needs to be explicitly stated.
11. For the FANS study, the authors need to provide the fluorescence dot plots to judge the quality of the staining/sorting.
12. One issue with this method is that burst/broken nuclei can appear to be TDP-43 dim. This may be occurring here as there seems to be a global downregulation of transcripts.
13. The validation of the FANS material is primarily based on assessment of differential polyadenylation that has recently been linked to loss of nuclear TDP-43. However, the finding of differential polyadenylation has been highly variable between research groups, and appears to be highly cell specific. Thus, more robust validation of proper sorting requires more strongly established methods such as evaluation of cryptic splicing (which admittedly is very difficult to do using single cell transcriptomics due to its 3' bias). If too sparse, expression of cryptically spliced genes would be helpful. Comparison of their FANS data with the existing FANS dataset which has been highly validated by many groups would also be helpful even if these were from different brain regions as there is likely to be high overlap between different brain regions.
14. Figure 6J and 6K suggests that loss of nuclear TDP-43 shows no selectivity for any cell type which runs counter to what is known about TDP-43 pathology.
15. Von Economo neurons are not present in the dorsolateral prefrontal cortex (page 16). It is also stated that TDP-43 pathology is found in cells of all layers (page 16) which needs a citation. I am not sure if this has formally been demonstrated, with pathologists characterizing TDP-43 pathology as present in superficial and deep layers with a predilection towards superficial layers. I do not believe that molecular demonstration that TDP-43 pathology affects all neuron subtypes has been done.

16. There is a focus in various parts on Von Economo neurons. However, it is known that TDP-43 pathology in most cases actually preferentially involves superficial cortical layers. The Seeley group has highlighted that VENs are also selectively vulnerable in FTLD-TDP (although not in other forms of FTD). A more balanced presentation of selectively vulnerable cell types would be helpful.

17. Figure 7, I was not sure what “case” refers to in the circular track. I understand that it refers to ALS versus ALS/FTD, but what does it mean that a certain cell type is associated with one group or the other? While I can understand, for example, that the oligodendrocyte is an “ALS-FTD” cell type with a certain amount of DEGs in ALS-FTD, what if a cell type shows significant DEGs in both ALS and ALS-FTD? How is one group selected over the other and why not show the data for both groups?

18. The circular track is also somewhat misleading in that the color scales seem arbitrary and may be exaggerating findings that were chosen to be highlighted, with some colors being binary versus graded. The choice of cutoff values for the various scales is not well described or justified.

19. The pathology in supplementary figure1 is of poor quality and not convincing.

Minor comments:

Page 14, line 330, I believe this should be Figure 2d (not 2b).

Page 18, line 429, it probably should read “We identified 67 DEGs in ALS”

(Remarks on code availability)

While I did not review the actual code, I did review their statistical methods which are well described.

Reviewer #3

(Remarks to the Author)

Overall, these datasets have the potential to be useful to the ALS/FTD research community.

While this is not the first study to combine snRNA-seq and snATAC-seq in ALS-FTD patient tissue (ref 47), the addition of FANS-snRNA-seq of a subset of samples allows for a novel comparison between the transcriptional signature of TDP-43-depleted neuronal nuclei with other cells in the motor cortex of ALS or ALS-FTD patients. Indeed, the FANS-snRNA-seq dataset alone is likely to be a useful resource, complementing the highly cited dataset generated by Liu et al (ref 10).

While these datasets will likely be useful resources, the authors should provide more context to describe the biological significance of their results. This is especially important given two counterintuitive results from this study: 1) a lack of cell type reductions in ALS or ALS-FTD motor cortex, and 2) convergent gene expression changes across varied cell types. These topics are addressed by the authors in the Discussion, but still warrant further investigation to ensure that these are true biological outcomes and not driven by technical artifacts.

Other key points that the authors must address to resolve concerning issues in the manuscript:

1) The abstract should be more detailed to highlight the key findings. The authors refer to a “well-defined transcriptomic signature of disease,” but no details/highlights are provided.

2) The authors control for various possible batch effects related to nuclear isolation and library prep. Were batch effects also considered between the 6 cohorts of postmortem tissue? This is very briefly assessed in Figure 2d in terms of cell-type proportions between groups, but other possible batch effects between cohorts are not mentioned.

3) As a related point, post-mortem interval (PMI) varied widely, from ~4 hours to 96 hours. Can any batch effects (or outliers) be explained by a transcriptional signature based on where the samples were collected/stored and/or by differences in PMI?

4) As the authors note, it is surprising that no cell types were robustly reduced in ALS or ALS/FTD samples, as these diseases are characterized by the death of neurons. The same brain region was assessed for all samples - because neuronal loss in this region could correlate with the location of symptom onset (bulbar vs spinal), have the authors considered this variable in their analyses?

5) Similarly, it is hard to interpret the fact that gene expression changes are convergent across cell types. This may be an important, though surprising, biological insight; however, the expectation would be that some cells (e.g., excitatory neurons) are primarily affected by disease while other cells (e.g., astrocytes, oligodendrocytes) undergo distinct transcriptional changes based on their unique cellular functions. This point warrants a deeper explanation – and careful consideration for any technical artifacts (or batch effects) that explain this result.

6) Given the importance of TDP-43 function in ALS/FTD, I do not think it's sufficient to say that splicing analysis and analysis

of cryptic exons are outside the scope of this study. There should be an effort to characterize alternative splicing events and particularly CEs known to be regulated by TDP-43. A recent paper (Gittings et al, PMID: 37466726) found that cryptic exons could be detected in snRNA-seq data from C9-ALS/FTD frontal cortex. Admittedly, this was most apparent in one sample that was deeply sequenced. However, assessing TDP-43 regulated splicing events in these data would provide another layer of information to compare with differential gene expression or chromatin accessibility between groups.

7) In Figures 3 and 4, the authors mention that they identified transcriptional signatures common to previous analyses and known to be associated with ALS, but more detail would be helpful since the Figures do not highlight specific DEGs. Highly relevant findings (STMN2, etc.) would be useful to mention specifically.

8) I do not see a supplementary file that includes DEGs from the FANS-snRNA-seq analysis. Please include this in any future submissions.

9) For the FANS-snRNA-seq experiment, approximately 25% of nuclei sorted based on NeuN/RBFOX3 expression were characterized as 'TDP-43-low.' This far exceeds the expected % of cells exhibiting TDP-43 pathology in ALS-FTD motor cortex. This warrants explanation, as it is a major discrepancy and undermines the veracity of the entire FANS-snRNA-seq data set.

10) The TDP-43 low population also has a different proportion of cell types, with over-representation of LINC00507 excitatory neurons. Using snRNA-seq data from healthy controls, do these neurons (or a subset of these neurons) normally have lower TDP-43 expression? An additional control is necessary to ensure that the DEGs associated with TDP-43 depletion are not driven by differences in cell type proportion between the TDP-43 high and low groups. Could this be modeled by doing DEG analysis on two sets of randomly sampled nuclei comprised of the same cell type proportions as the TDP-43 high/low groups?

Other important but less substantive issues that must be addressed are as follows:

11) Overall, I noticed many small typos while reading the paper. I would recommend carefully re-reading the text for small typos. Examples:

- o Fig S4 and S5,6: 'it parent cell population,' 'better suitable'
- o Page 17: 'randomly samples genes'
- o Figure 3: 'annotation of the sport'

12) The supplementary tables are not clearly labeled. If the file names are set by journal guidelines, the authors should include a brief description of the data within each file. For example, Suppl Tables 6-8 contain DEGs from pseudobulk analyses, but it is unclear which groups are being compared in each Table. All tables should include a description of the data within the file if the file names cannot be fully descriptive.

13) Methods: FANS was performed "as previously described" and a link is provided, but I do not see a citation.

14) Results: patients were "diagnosed based on standardized criteria (methods)" but I do not see this information in the Methods section. This is a minor point because these samples do have confirmed TDP-43 pathology in the motor cortex.

15) Demographics information is limited to age and sex. Is information on patient race/ethnicity available?

16) The organization of Figure 2 should be improved. 2b and 2c blend into one another. Additionally, the Figure 2d legend refers to the data as %, but the figure represents the data as proportions (i.e., 0.25 versus 25%).

17) Fig S5, S6, and S7 all have the same title. The authors should add details to define each figure (RNA-seq, ATAC-seq, etc.).

18) Page 24: "oligodendrocytes are affected by TDP-43 pathology in ALS-FTD, but not ALS" – this needs a citation.

19) Figure S10 G, H – the scale bars are set to highlight a lack of differences between C9 and sALS... but it seems that there are a few DEGs/DARs. It would be easier to interpret these panels if the axes were set lower (e.g., +/- 500).

20) Figure S11 uses "HC" as healthy control instead of Ctrl like other figures.

21) Figure S11 "Increasing dot size represents higher (red) or lower (blue) normalized enrichment score ('NES') for the respective term" - It is unclear to me if this is the same as saying blue dots represent downregulated genes, while red dots represent upregulated genes. If not, it would be more intuitive to see what pathways are enriched for up- vs down-regulated genes.

(Remarks on code availability)

This is not an area of expertise for me, so should be evaluated by someone more knowledgeable in computation and bioinformatics.

Reviewer #4

(Remarks to the Author)

The authors used multi-modal single-nucleus profiling in the primary motor cortex of ALS and ALS-FTD to understand gene signatures and epigenetic regulatory mechanisms. The novelty of the study lies in their discovery of a distinct subset of transcriptomic changes not linked to TDP-43 pathology. Extensive bioinformatic data analyses were performed, providing a valuable single-nucleus dataset for the ALS field. However, the lack of a detailed molecular mechanism evaluation is a weak point of this study. The study could be significantly improved by performing experimental validation assays to support their hypothesis. Currently, no detail evaluation has been conducted for their findings.

(Remarks on code availability)

Version 1:

Reviewer comments:

Reviewer #1

(Remarks to the Author)

I am pleased to see that the authors have impressively strengthened their manuscript with the addition of the cryptic exon analysis and the deeper dive into the FANS-seq data. I have no further criticisms, though I implore the authors to make sure that all sequencing data, metadata, and count matrices are made publicly available at the point of publication.

(Remarks on code availability)

The code is exceptionally well organized. I congratulate the authors for being so transparent.

Reviewer #2

(Remarks to the Author)

I greatly appreciate the author's willingness to address the many comments I made in the first round of reviews. This revised manuscript is greatly improved and I have a lot more confidence in the findings. Significant improvements include comparison of their results with the prior studies, and improved presentation and validation of their FANS dataset. I only have a few minor comments.

1. The "enrichment" of microglia in superficial layer 1 is interesting, but I would consider that this might not be an increase in superficial microglia but rather a decrease in other cellular elements. Given the paucicellular nature of layer 1, it is not surprising that other cell types are reduced which overall would appear like an increase of microglia. However, histologically, there is no accumulation of microglia in layer 1.

2. I appreciate the wider maps shown in supplementary figure 13 for the rare Ecx FEZF NTNG1 cells. However, in some panels, there are dots that seem to indicate these are also found in superficial layers. This is likely due to the fact that different color scales are used for each pane (ranging from 0 to 6 versus 0 to 3). I suggest making all the panels use the same color scale, even if that means some panels do not identify these rare cells (as expected).

3. The legend for supplemental figure #1 needs to be updated to include a description of panels e to k.

4. I appreciate that the FANS dot plots are now shown. For the gating, it appears that DAPI singlets and doublets were included. If this is true, I would adjust the description in the methods where it says that only "single nuclei" were obtained.

5. There is a typo on page 29 where "find" should be "found"

(Remarks on code availability)

Reviewer #3

(Remarks to the Author)

The authors have provided substantive updates and clarifications to the original manuscript, which will make it more useful to the ALS research community.

(Remarks on code availability)

We are grateful to the Reviewers for their effort and constructive criticism, which helped us to improve the manuscript. We have thoroughly revised the study, including a significant amount of new data and new experiments. Below is the point-to-point response to each issue raised. We thank the Reviewers again.

Reviewer #1:

Reviewer #1 (Remarks to the Author):

The authors have done a very thorough job in generating and describing singlenucleus multi-omic data from the motor cortex of ALS and ALS-FTD patients, alongside FANS sorted single-nucleus RNA-seq to identify nuclei with low TDP-43. Both these datasets will be tremendously useful to the wider field. All analyses have been done carefully and using the current state of the art tooling. I have only a few suggestions.

Given that the authors generated genotyping data to aid with demultiplexing of the single-nucleus data, it was a shame that this data wasn't incorporated into the paper, for example by looking for ALS-associated common variants that overlapped their chromatin accessibility regions, or even mapping QTLs if the sample size allows. Is this something the authors would consider?

Response to Reviewer #1:

Point 1: *Given that the authors generated genotyping data to aid with demultiplexing of the single-nucleus data, it was a shame that this data wasn't incorporated into the paper, for example by looking for ALS-associated common variants that overlapped their chromatin*

accessibility regions, or even mapping QTLs if the sample size allows. Is this something the authors would consider?

Reply: Following the reviewer's suggestion we analyzed ALS-associated common variants that overlapped in chromatin accessible regions. We found 24 unique SNPs from two list of GWAS SNPs associated to ALS (222 total SNPs) overlapping chromatin accessibility peaks. We investigated the chromatin accessibility of these peaks and did not observe a significant differential accessibility in ALS or ALS-FTD. We added these data as new Suppl. Fig. 18 (Results section 'Chromatin remodeling...', L 651-658).

Point 2: *The enrichment of neurodegenerative disease KEGG pathways in the DEGs used throughout the manuscript is likely due to all of those "disease" pathways having strong overlaps with the genes in the Oxidative Phosphorylation KEGG pathway. I would suggest the authors either remove these assertions, or refine it by testing only the genes that are unique to each disease pathway. Alternatively, the authors could perform enrichment with lists of GWAS and rare variant-linked genes for ALS.*

Reply: The Reviewer is right that disease pathways have strong overlaps between each other and with genes in the Oxidative Phosphorylation KEGG pathway. We thank for this useful remark! We analyzed in detail the enrichment and found that most of the aforementioned pathways are enriched just by the fact that gene expression comes from the (motor) cortex, whose genes are heavily annotated to AD, PD, ALS, etc. Using only genes expressed in the motor cortex as a background alleviates this problem. We have added to the manuscript this important observation (Methods section 'Functional analysis', L 336-339; new Suppl. Fig. 10; Results section 'The ALS/ALS-FTD motor cortex is transcriptionally altered...', L 575-578).

Point 3: *Were the authors able to assess cryptic splicing of STMN2 and KALRN in the TDP-43 low fraction of single cells, given that those two genes have been previously demonstrated to be detectable in single cell data? PMID: 37466726.*

Reply: We thank the Reviewer for this comment since cryptic exons are increasingly recognized as highly relevant in ALS. In the revised manuscript, we have incorporated cryptic exon detection as an integral part of the analysis. We were able to detect the cryptic exons in *STMN2* and *KALRN* in the TDP-43^{Low} fraction. We have added this data to the revised manuscript (new Fig. 7 in all TDP-43^{Low} cells & Fig. 8 in a cell-type specific manner of TDP-43^{Low} cells ; Results sections ‘Nuclear TDP-43 pathology selectively affects...’ and ‘Transcriptional effects of TDP-43 pathology...’, L 719-724 & L 738-740).

Point 4: *Similarly, did the authors investigate whether NPTX2 was upregulated in the TDP-43 low fraction, given its recent demonstration as a TDP-43 target? PMID: 38355792*

Reply: We followed this suggestion and investigated *NPTX2* expression in our data. Indeed, we found a significant down-regulation of *NPTX2* in the TDP-43^{Low} fraction and, interestingly, of its receptor *NPTXR*. Furthermore, we found a downregulation of *NPTX2* also in the multi-omic data set regarding Exc-RORB (L3-5 ITC) and Exc-LINC00507 (L2/3 ITC) neurons in ALS and Exc-THEMIS (L6 ITC) and Exc-LINC00507 (L2/3 ITC) neurons in ALS-FTD. We have included this analysis in the revised manuscript (Results section ‘The ALS/ALS-FTD motor cortex is transcriptionally altered...’, L 577 & L 613, Results section ‘Nuclear TDP-43 pathology selectively affects’, L 717-719).

Point 5: *Please use commas as thousand separators throughout.*

Reply: As suggested we added commas as thousand separators in the whole manuscript including figures and supplemental figures.

Point 6: *Line 42 - STMN2 typo*

Reply: We thank the reviewer for his/her/their attention and corrected this typo (L 82).

Point 7: *Line 56 - the vast majority of polymorphisms detected by GWAS are in non-coding regions, I would change “some” to “most”.*

Reply: We thank the Reviewer. This part is not anymore a part of the ‘Introduction’ section.

Point 8: *Fig. 4 - typo, ALS-FTD has 17 donors, not 30*

Reply: We apologize for this typo and corrected it (Fig. 4).

Reviewer #2 (Remarks to the Author):

This manuscript describes concurrent RNA and chromatin accessibility sequencing of single nuclei from ALS, ALS-FTD and control motor cortex. This dataset emerges as a few other single cell datasets have been published, although this is the first using the technology that allows for both transcriptome and epigenome profiling in the same cell. The analysis pipeline is sophisticated, and a large number of cases are studies. However, the results are sometimes inconsistent with what is known which may be driven by relatively low coverage and/or cells per case. Moreover, while this is an impressive amount of data and analysis, I am not sure what additional insights these analyses have put forward towards our understanding of ALS and ALS/FTD. These issues include the following:

1. The cases/demographics file is not well formatted and is incomplete. In addition to demographic information, the reader needs information on the number of cells, number of reads per cell, number of cell types, etc. on a per case basis. While it is impressive that there is a large number of cases that were included, it does not seem that each case was sequenced to equal depth when examining figure 2d. Based on the number of cases and the number of total nuclei, the average number of cells per has is on the order of 2000-2500, with many cases likely being much lower, as evidenced by figure 4a where some cases have no

nuclei from some cellular subtypes. This is problematic because some of the analyses are likely underpowered if there are not sufficient numbers of cells per case (and the number of cells per case is used as a variable in some analyses).

Point in the summary: *The analysis pipeline is sophisticated, and a large number of cases are studied. However, the results are sometimes inconsistent with what is known which may be driven by relatively low coverage and/or cells per case.*

Reply: We have now dedicated a dedicated part of the revised manuscript to the comparison to the previous studies (L486-494). We show that both our multi-omic dataset is comparable by major quality criteria to the three largest previous studies, including by coverage. We further demonstrate that the cell-type identification and the main findings are strongly concordant with the previous studies: most changes in upper layer excitatory neurons, concordant changes across excitatory neurons, no significant effects by the C9ORF72 HRE genotype, more pronounced differences in ALS-FTD, oligodendrocyte and astrocyte component. To address the concern with low number of cell per case, we have performed additional analyses (see point # by Reviewer #2).

Moreover, while this is an impressive amount of data and analysis, I am not sure what additional insights these analyses have put forward towards our understanding of ALS and ALS/FTD.

Reply: We have explicitly highlighted the additional insights that our study delivers in the revised manuscript. The most important are:

- With the new experimental data, we are the first to analyze the cell-type specific effects of TDP-43 pathology in the ALS/FTD patients' motor cortex. Most important, we show that the previous extensive report (Liu et al.) delivers mostly markers for affected cell types, and not genes which are affected by TDP-43 pathology in these cell types.

- We define for a first time with high resolution which subtypes of neurons are affected by TDP-43 pathology in the ALS/ALS-FTD motor cortex
- We deliver novel sets of TDP-43 pathology-affected genes in the 5 mostly affected cell subtypes
- All these findings were enabled by our multi-omic approach, which allowed our study to be the first to reach such high cell-type resolution (Fig. 1)
- We demonstrate that TDP-43 pathology-related transcriptomic changes make up only a small part of the total transcriptomic changes observed in the ALS/ALS-FTD motor cortex

Point 1. *The cases/demographics file is not well formatted and is incomplete. In addition to demographic information, the reader needs information on the number of cells, number of reads per cell, number of cell types, etc. on a per case basis. While it is impressive that there is a large number of cases that were included, it does not seem that each case was sequenced to equal depth when examining figure 2d. Based on the number of cases and the number of total nuclei, the average number of cells per has is on the order of 2000-2500, with many cases likely being much lower, as evidenced by figure 4a where some cases have no nuclei from some cellular subtypes. This is problematic because some of the analyses are likely underpowered if there are not sufficient numbers of cells per case (and the number of cells per case is used as a variable in some analyses).*

Reply: We thank the Reviewer and have addressed this point. We have provided the requested data in the new Suppl. Tables 22-25.

We had considered the number of cells per donor already in the original report. For cell-type abundance analyses, we have incorporated this variable as covariate in the mixed-effects linear model MASC. For differential gene expression analysis, we have used pseudo-bulk approach only down to the level of major cell types. Du to the very low and highly variable

number of cells per donor at the highest resolution level, 'cell types/subtypes', we have employed MAST for DGE analysis on this level.

Point 2. *The lack of differences in proportion of various cell types in disease versus control is worrisome that this dataset is not capturing the molecular events in disease. This may be related to the low number of cells per case (point #1).*

Reply: We thank the Reviewer for this point, which helped to improve the quality of the study. We dedicated an extensive investigation to this observation. In sum, we demonstrate that the lack of differences in proportions *does* agree with previous – comparable – studies and is attributable to experimental noise due to the nature of the measurement. This and the fact that we recapitulate previously established transcriptomic changes (downregulation of *STMN2*, of *NPTX2*, inclusion of cryptic exons, changes in the same cell types as previously described, etc.) demonstrate that our study **does** faithfully capture the molecular events of the disease.

Three lines of evidence suggest that the lack of cell-type differences in our study are not indicative of poor quality of the dataset or 'not capturing the molecular events in disease':

- 1) Very careful modeling and control for bias (see below)
- 2) Same findings by the previous studies (please see table below)
- 3) Previously described phenomenon that single-cell/nucleus RNA-seq often doesn't reflect IHC censuses (e.g. Pineda et al.)

Detailed description of each argument 1-3:

1) We have shown in our analysis that there's no statistically robust depletion or expansion of any specific cell type except in some ALS or ALS-FTD samples using plain

statistics (repeated Wilcox test), and using sophisticated modeling with three different frameworks:

- Deep-learning aided Bayesian statistical modelling with CODA,
- Hierarchical analysis with tascCODA
- Linear mixed models with random and fixed effects.

Moreover, we repeated the analysis over three hierarchical levels of cell-type identification. To address the Reviewer's concern, we also repeated the analysis only considering samples with high number of cells (>2000). None of the analyses showed a significant change in any cell type in ALS, ALS-FTD or C9-ALS-FTD, with the exception of Astrocytes and L3-5 ITC Exc neurons in less stringent analyses. We added these results as the new Suppl. Fig. 11.

2) Our observation is in accordance with 5 previous studies of C9-ALS/ALS-FTD motor cortex. None of the 5 studies (see table below) found loss of specific neuronal cell types in ALS/ALS-FTD *in the motor cortex with snRNA-seq*. Please note that changes were found often in the DLPF or frontal cortex, and more often in FTD; but not in ALS/ALS-FTD in the motor cortex, what we have in our study. Moreover, Li et al., a very-well controlled study, also found no reduction of any cell type in the motor cortex of ALS and FTD not only by single-cell compositional analysis, but also with flow cytometry. Our findings thus do not surprise, but re-affirm previous analyses.

For-review-table 1. Previous observations of cell-type proportions in comparable studies. Where authors found differences in cell type abundance, we have highlighted in red the difference to our study.

Study	Disease group	Brain area	Method	Finding
Li et al. 2023	C9-ALS, C9-ALS-FTD	BA4 motor cortex	snRNA-Seq	No significant changes

Li et al. 2023	C9-ALS, C9-ALS-FTD	BA4 motor cortex	Flow cytometry of same samples	No significant changes
Gittings et al. 2025	C9-ALS, C9-ALS-FTD & C9-FTD	Frontal cortex	snRNA-Seq	L5 ETC neurons reduced only in C9-FTD , but not C9-ALS or C9-ALS-FTD
Limone et al. 2022	sALS	BA4 motor cortex	snRNA-Seq	No significant changes
Wang et al. 2025	C9 ALS-FTD	BA9 DLPF cortex	snRNA-Seq	Decrease in Exc neurons
Pineda et al. 2024	C9 ALS, C9-ALS-FTD, sALS, sALS-FTD	BA4 motor cortex & BA9 DLPF cortex	snRNA-Seq	Reduction of ETC neurons only in DLPF cortex , but not motor
Pineda et al. 2024	C9 ALS, C9-ALS-FTD, sALS, sALS-FTD	BA4 motor cortex & BA9 DLPF cortex	IHC/IFC	Reduction of ETC neurons in motor cortex (ALS) and DLPF cortex (ALS-FTD)

3) Previous studies have described a pronounced discrepancy between single-cell/nucleus compositional data and IHC/flow cytometry censuses (Li et al., Pineda et al.). Possible factors include the nature of the single-cell/nucleus data (compositional) and the way a cell-type identity is determined in these experiments. Indeed, Pineda et al. demonstrated a marked loss of L5b ETC neurons in ALS motor cortex with IHC, despite no differences in the single-nucleus RNA-seq dataset from the same samples. The authors of this study themselves conclude in their 'Limitations of the study' section: *“Another limitation of the assay and use of human tissues is the expected variability in dissections and nuclei capture resulting in highly variable cell type yields even from high quality and rigorously dissected tissues, making it a poor method for evaluating changes in cell type*

composition. This mandates the use of histology, or similar approaches, to validate such changes”

In sum, we acknowledge that this question is of central importance and have improved the text accordingly: (Result section “Hierarchical annotation of cell-type identity...”, L 495-507 & Discussion L 828-835). We have followed the Reviewer’s suggestion and checked with high-cell number samples only. All new data listed above is included in the new Suppl. Fig. 11 and Suppl. Fig. 15. We will also be happy to include the comparison to the literature (table above) in the revised manuscript, but such data is usually out of the scope of research articles and better fits a systematic review article.

Point 3. *Figure 2A contains the data shown in figure 1D but broken up into different parts of the figure. As this data is repeated and is not really contributing additional data to figure 2, and since the breaking up of the UMAP further distorts the UMAP which is already non-linear, I suggest not including these dot plots.*

Reply: We thank the Reviewer for this recommendation and acknowledge that this UMAP representation is redundant and had just illustrative value, as was indicated in the original text. We have removed these parts of the figure and have added a Bonsai representation of the data to aid the reader in comprehending the structure of the data (please see modified Fig. 2).

Point 4. *Page 18 lines 438-439 states that the data was compared to “previous studies” which I do not see in the supplementary figure 11 – unless they mean just mapping their data using GSEA to KEGG pathways. This should be clarified.*

Reply. *We have clarified this issue and the text (Results section “Transcriptomic remodeling of the ALS.ALS-FTD motor cortex, L 571-572). For the comparison to previous studies, see next point by Reviewer #2.*

Point 5. *As two large single cell datasets have been published in the last couple of years on this topic, a comparison of findings across datasets would be important to show either consistent (or discordant) results.*

Reply: We have acknowledged the Reviewer's suggestion and are very thankful for it, as it helped us to significantly improve the manuscript.

Our study robustly agrees with previous findings: similar changes between major cell types, more pronounced differences between phenotypes, most changes in L2/3 and L5/6 neurons, dysregulation in oligodendrocytes in ALS-FTD, and similar differentially expressed genes when same analysis applied (e.g. the TDP-43 low pseudobulk analysis). In particular, our results are mostly in agreement on these points with Li et al. and Pineda et al., which were very comprehensive and robust studies. We have added these comparisons to the revised manuscript (L495-497, L615-619, L828-835). We also compared the FANS-seq dataset to the two previous report most similar to it: Liu et al. and Gittings et al. (L716-725).

Point 6. *Figure 3 is problematic and appears to show that this analysis pipeline, matching single cell transcriptomes to existing spatial transcriptomic data, is not working properly. RORB is known to be a layer 4 gene, but they report it highest in layer 5. Similar problems exist for Themis (layer 5, not 6), FEZF2 (layer 5, not 6), SST (layers 2,3,5, not layer 1), microglia (equally dispersed throughout the CNS). This is important because the same methods are used later to map the authors' new data – and so if the method cannot do this accurately for some known genes (such as Figure 4N), it casts doubt on the new data localization.*

Reply: We are very grateful to the Reviewer for this constructive critique, which made us realize that the naming of the major transcriptomic cell types may be confusing to some readers. We have therefore thoroughly revised the naming scheme for the cell types, which are now described with their localization/morphology, similar to other reports in the field (Gittings et al.). Indeed, Fig. 3 does **not** demonstrate the expression of the *RORB*, *THEMIS*

and *FEZF2* **genes** in different cortical layers, **but of the cell classes** which are named after these markers (although these markers are not exclusively expressed only in one class).

Interestingly, as the Reviewer remarks, the spatial data demonstrates an *unexpected* enrichment of microglia in L1 of the motor cortex. While the enrichment of microglia in the white matter relative to the grey matter is well-known, we were not aware that the enrichment of microglia in L1 has not been described so far. Given that the LIBD dataset is carefully controlled and highly validated, we have added this important information to the revised manuscript (Results section 'Spatial transcriptomics allow for the mapping', L 538-539).

Point 7. *Figure 3f shows a single dot in a very small inset as evidence of localization of Betz cells. This is not convincing.*

Reply: We understand the Reviewer's skepticism and have made an effort to demonstrate the identification of the Betz cells more convincingly. In the new Supp. Fig. 9 n, we show that Exc FEZF2 NTNG1 cells overlap almost perfectly with the annotations of Betz cells from three different previous studies, thus unequivocally demonstrating their identity as L5 ETC neurons. We have also added additional spatial transcriptomics images to demonstrate the correct localization of the transcriptomic correlate of Betz cells in the new Suppl. Fig. 13. Importantly, as ETN markers are very specific for Betz cells, it is expected that they are expressed very concentrated and intensive in single spots that lie beneath Betz cells. Furthermore, as they are very rare (< 0.4 percent in the multiome dataset), they are expected to be very sparse in the cross-sectional images of the spatial data. We have specified this in the revised text (Results section 'Spatial transcriptomics allow for the mapping...', L 544-550).

Point 8. *The Venn Diagram in figure 4N, while significantly “enriched” in terms of overlap (likely due to the high statistical power when compared against the entire transcriptome), is striking for how the vast majority of the results do not overlap.*

Reply: This is indeed an interesting observation. We have characterized it further in the new Suppl. Fig. 16, 17 to show that many of the non-overlapping genes are regulated in the same manner (direction, magnitude), but do not reach significance in one of the groups (ALS or ALS-FTD), suggesting that the changes are more similar than it seems, but the strength of dysregulation varies between the both groups. Still, the changes are less concordant between the disease groups (e.g. L2/3 ITC nuclei ALS vs ALS-FTD) than between the cell types in the same disease group (e.g. L2/3 ITC nuclei vs L3-5 ITC nuclei, both in ALS), suggesting a ‘primarily disease-specific’ and secondary ‘cell-type specific’ changes. This is in strong agreement with the observations of Pineda et al. (Fig. 1 A “4” and “5” in Pineda et al.) and Li et al. (concluding remarks in ‘Results’ in Li et al.). We have added this valuable insight to the revised manuscript (Results section ‘The ALS/ALS-FTD motor cortex is transcriptionally altered’, L 614-623).

Point 9. *Page 21, line 508, states that DARs “correlated strongly”. While some p-values are very low due to the high number of datapoints, the r values (which is used to judge the strength of a correlation) are very low suggestive of a very weak correlation.*

Reply: The Reviewer refers to Fig 5 h, but the text mentioned comments Fig. 5 I, where r-values are in the range 0.7-0.87, which is strong correlation. We have refined the text to explicitly reference subpanel “5 i” to make clear which data does this statement describe (Results section ‘Chromatin remodeling is associated with a subset’, L 647-651).

Point 10. *Figure 5i, can the authors clarify on which dataset the Spearman correlations were performed? It appears to be just for the data that is both differentially expressed and*

differentially accessible which is fine but just needs to be explicitly stated.

Reply: As correctly observed by the Reviewer, the correlations are performed on both differentially expressed and differentially accessible datapoints only. We have specified this explicitly (Results section 'Chromatin remodeling is associated with a subset', L 647-651). We thank the Reviewer for pointing this out.

Point 11. *For the FANS study, the authors need to provide the fluorescence dot plots to judge the quality of the staining/sorting.*

Reply: The Reviewer is absolutely right about this. We have provided the fluorescence dot plots along with percentages in the new Fig.6 e-g and in the new Suppl. Fig. 20.

Point 12. *One issue with this method is that burst/broken nuclei can appear to be TDP-43 dim. This may be occurring here as there seems to be a global downregulation of transcripts.*

Reply: We thank the Reviewer for this critical remark. It is indeed very important to control that TDP-43^{Low} nuclei are not just burst/broken nuclei. We have added a substantial new amount of data to the study to increase the quality of the FANS-seq dataset. We have several lines of strong evidence that the TDP-43^{Low} nuclei are not just burst/broken:

- 1) The strongest evidence comes from our analysis of gene expression and RNA processing in TPD-43^{Low} vs. TDP-43^{High} nuclei. Not only could we reproduce previously validated transcription programs of TDP-43 pathology, but also specific alternative poly-adenylation targets (*MARK3*, *ELP1*) and the inclusion of cryptic exons specific for TDP-43 pathology (*STMN2*, *KALRN*). Thus, it is highly unlikely that the TDP43^{Low} fraction of sorted nuclei is mainly debris. All these data is demonstrated in the new Fig. 7.
- 2) The gating strategy in the fluorescence-activated nuclei sorting was designed such that not only typical debris, but also all putative debris was not selected (Suppl. Fig. 20, first

gate). Furthermore, nuclei were positively selected for uniform signal (the singlet selection) and for DAPI, which would have had a reduced signal due to DNA release if nuclei were ruptured (Suppl. Fig. 20 2nd, 3rd and 4th gate). This strategy is now demonstrated in the new Suppl. Fig. 20.

3) The gating/selection of TDP-43^{High} against TDP-43^{Low} nuclei was designed to be relative to the NeuN fluorescence signal (please see new Fig. 6, panel 'e': the selection gates for TDP-43^{High} and TDP-43^{Low} are 'diagonal' instead of 'horizontal' meaning that nuclei were only sorted as "TDP-43^{Low}" if their TDP-43 signal was *proportionally* lower (i.e. lower than expected due to the strength of their NeuN signal). Furthermore, Fig. 6 e demonstrates that a lot of nuclei with poorer signal were sorted to "TDP-43^{Low}" as well, so no bias is expected.

4) Next, we acknowledge that the global down-regulation of transcripts (including non-significant) was problematic. We have inspected its roots and found out it's a combination of the lower sequencing depth of TDP-43^{Low} nuclei and of the implementation of L2FC in Li et al. ¹, which we used. We performed control DGE analyses with equal numbers of nuclei or counts and found out this implementation is sensitive to different library sizes (This was not a problem in Li et al., since they had comparable library size between the groups they compared). To overcome this problem, we increased the overall sequencing depth of the dataset; we performed further FACS experiments and increased the sample size of the dataset, and, finally, proceeded to using the MAST model fold change estimates, which in contrast to the implementation of Li et al. is not sensitive to lower library sizes in one of the groups. We specifically thank the Reviewer for this remark, as this was a very important aspect to control for bias and would have invalidated the conclusions about pathological TDP-43 cells and its consequence on transcriptomics.

Point 13. *The validation of the FANS material is primarily based on assessment of differential polyadenylation that has recently been linked to loss of nuclear TDP-43. However, the finding of differential polyadenylation has been highly variable between*

research groups, and appears to be highly cell specific. Thus, more robust validation of proper sorting requires more strongly established methods such as evaluation of cryptic splicing (which admittedly is very difficult to do using single cell transcriptomics due to its 3' bias). If too sparse, expression of cryptically spliced genes would be helpful. Comparison of their FANS data with the existing FANS dataset which has been highly validated by many groups would also be helpful even if these were from different brain regions as there is likely to be high overlap between different brain regions.

Reply: We thank the Reviewer for these constructive suggestions and have followed both of them. The Reviewer is right that cryptic exons in *STMN2* and *KALRN* are only included in TDP-43 pathology, while APA may just reflect the differences in cell types between TDP-43^{Low} and TDP-43^{High} nuclei. We have performed further rounds of FANS-seq and more rounds of sequencing to increase the size, quality and sequencing depth of the FANS-seq dataset. The increased quality of the dataset allowed us to reliably quantify the inclusion of known TDP-43-specific cryptic exons. We also compared our data to Liu et al. and observed very concordant results. All these new data is provided in the new Fig 7.

Point 14. *Figure 6J and 6K suggests that loss of nuclear TDP-43 shows no selectivity for any cell type which runs counter to what is known about TDP-43 pathology.*

Reply: The Reviewer is correct with the expectation that TDP-43 pathology should be selective for cell types. The new, improved FANS-seq dataset allowed us to indeed demonstrate this: TDP-43^{Low} nuclei are selectively enriched for L2-L3 intratelencephalic neurons, as well as for L5-6 near-projecting, extratelencephalic and corticothalamic, and intratelencephalic neurons. TDP-43 pathology is specific for specific subtypes like Exc LINC00507 *FREM3*, Exc THEMIS LINC00343 and Exc FEZF2 *NTNG1* and can be demonstrated not only by their enrichment in the TDP-43^{Low} fraction, but also by strong transcriptional dysregulation and inclusion of cryptic exons in these cell types. These new data are central to the conclusions of the revised manuscript and are included in the new

Fig. 6-9 (new Results sections 'Nuclear TDP-43 pathology selectively affects excitatory' and 'Transcriptional effects of TDP-43 pathology are cell-type specific', L659-754).

Point 15. Von Economo neurons are not present in the dorsolateral prefrontal cortex (page 16). It is also stated that TDP-43 pathology is found in cells of all layers (page 16) which needs a citation. I am not sure if this has formally been demonstrated, with pathologists characterizing TDP-43 pathology as present in superficial and deep layers with a predilection towards superficial layers. I do not believe that molecular demonstration that TDP-43 pathology affects all neuron subtypes has been done.

Reply: Indeed, Von Economo neurons are not generally expected in the DLPF cortex, even though some studies have demonstrated this ². Several pathological studies have demonstrated that TDP-43 pathology may affect different layers of the motor cortex ³. We have added these to the text (Results section 'Spatial transcriptomics allow for the mapping, L 519-520). The Reviewer is correct that, to the best of our knowledge, there have not been a molecular demonstration that TDP-43 pathology affects all neuron subtypes so far. We have addressed this issue with our new data, providing for a first time highly-resolved characterization of the transcriptional cell types affected by TDP-43 pathology. We have improved the text to specify more precisely that TPD-43 pathology affects all **excitatory** neuron types in the motor cortex (Results section 'Nuclear TDP-43 pathology selectively affects excitatory', L 694-699).

Point 16. *There is a focus in various parts on Von Economo neurons. However, it is known that TDP-43 pathology in most cases actually preferentially involves superficial cortical layers. The Seeley group has highlighted that VENs are also selectively vulnerable in FTLD-*

TDP (although not in other forms of FTD). A more balanced presentation of selectively vulnerable cell types would be helpful.

Reply: We agree with the Reviewer on this point and have revised the manuscript for a more balanced presentation of the selectively vulnerable cell types. Our new data clearly demonstrates that not only extratelencephalic neurons, but also intratelencephalic neurons from different cortical layers, mostly the superficial layers L2/3 are subject to TDP-43 pathology (Results section 'Nuclear TDP-43 pathology selectively affects excitatory', L 697-700).

Point 17. *Figure 7, I was not sure what “case” refers to in the circular track. I understand that it refers to ALS versus ALS/FTD, but what does it mean that a certain cell type is associated with one group or the other? While I can understand, for example, that the oligodendrocyte is an “ALS-FTD” cell type with a certain amount of DEGs in ALS-FTD, what if a cell type shows significant DEGs in both ALS and ALS-FTD? How is one group selected over the other and why not show the data for both groups?*

Reply: We understand that this data visualization can be challenging and have reduced the complexity of this figure. We also added new labels to make clear that each leave in the dendrogram represents the changes for a specific cell type in a specific disease group comparison (new Suppl. Fig. 17). We thank the Reviewer.

Point 18. *The circular track is also somewhat misleading in that the color scales seem arbitrary and may be exaggerating findings that were chosen to be highlighted, with some colors being binary versus graded. The choice of cutoff values for the various scales is not well described or justified.*

Reply: We removed the figure in question from the revised manuscript (see previous point), since we feel the critique is reasonable and the figure was not balanced enough. For the

record, the Reviewer is absolutely correct about this - the color schemes were chosen for better contrast to make the figure more readable, but this seems to be misleading to readers.

Point 19. *The pathology in supplementary figure 1 is of poor quality and not convincing.*

Reply: We agree with the Reviewer and provide new IHC pathology images from two different pathologists in the new Suppl. Fig. 1.

Minor points:

Page 14, line 330, I believe this should be Figure 2d (not 2b).

Thanks, we have corrected this inconsistency.

Page 18, line 429, it probably should read "We identified 67 DEGs in ALS"

We apologize for these inconsistency and have corrected it.

Reviewer #2 (Remarks on code availability): *While I did not review the actual code, I did review their statistical methods which are well described.*

Reply: We thank the Reviewer for acknowledging the quality of our analysis. We have added even more details to the description of the statistical methods in hope to provide a robust and reproducible analysis (please see changes marked in red in the revised 'Methods' section, e.g. L 357-380).

Reviewer #3 (Remarks to the Author):

Overall, these datasets have the potential to be useful to the ALS/FTD research

community.

While this is not the first study to combine snRNA-seq and snATAC-seq in ALS/FTD patient tissue (ref 47), the addition of FANS-snRNA-seq of a subset of samples allows for a novel comparison between the transcriptional signature of TDP-43-depleted neuronal nuclei with other cells in the motor cortex of ALS or ALS-FTD patients. Indeed, the FANS-snRNA-seq dataset alone is likely to be a useful resource, complementing the highly cited dataset generated by Liu et al (ref 10).

While these datasets will likely be useful resources, the authors should provide more context to describe the biological significance of their results. This is especially important given two counterintuitive results from this study: 1) a lack of cell type reductions in ALS or ALS-FTD motor cortex, and 2) convergent gene expression changes across varied cell types. These topics are addressed by the authors in the Discussion, but still warrant further investigation to ensure that these are true biological outcomes and not driven by technical artifacts.

Response to Reviewer #3:

Point in the summary. *While these datasets will likely be useful resources, the authors should provide more context to describe the biological significance of their results. This is especially important given two counterintuitive results from this study: 1) a lack of cell type reductions in ALS or ALS-FTD motor cortex*

Reply: We have extensively addressed this issue, which was raised also by Reviewer #2. See detailed reply to Point #2 by Reviewer #2. In short, we show that cell census is unprecise with this method and the lack of cell-type reduction agrees with the previous studies.

Point in the summary. *2) convergent gene expression changes across varied cell types. These topics are addressed by the authors in the Discussion, but still warrant further*

investigation to ensure that these are true biological outcomes and not driven by technical artifacts.

Reply: We have addressed this issue and added a new analysis in the revised manuscript (new Suppl. Fig. 16 & 17), as well as commented on its concordance with previous studies Results section 'The ALS/ALS-FTD motor cortex is transcriptionally altered', L 614-619).

Other key points that the authors must address to resolve concerning issues in the manuscript:

Point 1. *The abstract should be more detailed to highlight the key findings. The authors refer to a “well-defined transcriptomic signature of disease,” but no details/highlights are provided.*

Reply: We agree with the Reviewer. Since we thoroughly re-worked the study and got new findings, the abstract is now completely revised, taking care to highlight precisely the key findings ('Abstract', L 36-56).

Point 2. *The authors control for various possible batch effects related to nuclear isolation and library prep. Were batch effects also considered between the 6 cohorts of postmortem tissue? This is very briefly assessed in Figure 2d in terms of cell-type proportions between groups, but other possible batch effects between cohorts are not mentioned.*

Reply: This is a very important point and we have commented on it in the revised Methods section. Since we were unable to consider PMI, onset site, BMI and other possible batch effects as the data was incomplete, we used SVA analysis to account for hidden/unknown batch variables in the differential gene expression analysis. We have revised the manuscript

to explicitly state this (Methods section 'Differential expression/accessibility testing', L 314-317 and Results section 'The ALS/ALS-FTD motor cortex is transcriptionally altered', L 555-559).

Point 3. *As a related point, post-mortem interval (PMI) varied widely, from ~4 hours to 96 hours. Can any batch effects (or outliers) be explained by a transcriptional signature based on where the samples were collected/stored and/or by differences in PMI?*

Reply: This is correct and likely. Since we couldn't properly include PMI as a covariate, we modelled batch covariates with SVA (please see response to the previous point).

Point 4. *As the authors note, it is surprising that no cell types were robustly reduced in ALS or ALS/FTD samples, as these diseases are characterized by the death of neurons. The same brain region was assessed for all samples - because neuronal loss in this region could correlate with the location of symptom onset (bulbar vs spinal), have the authors considered this variable in their analyses?*

Reply: We thank the Reviewer for this useful comment. Unfortunately, we did not have enough onset data to stratify samples by it. We have explicitly highlighted this in the section about limitations of this study (Discussion section 'Limitations of the study', L 870-873).

Point 5. *Similarly, it is hard to interpret the fact that gene expression changes are convergent across cell types. This may be an important, though surprising, biological insight; however, the expectation would be that some cells (e.g., excitatory neurons) are primarily affected by disease while other cells (e.g., astrocytes, oligodendrocytes) undergo distinct transcriptional changes based on their unique cellular functions. This point warrants*

a deeper explanation – and careful consideration for any technical artifacts (or batch effects) that explain this result.

Reply: We agree with the Reviewer and have dedicated a part of the revised Results to this question. Indeed, convergent gene expression changes across cell types are somewhat surprising, although we also find divergent changes in glial cells. In the new Suppl. Fig. 17, we show that changes in glial cell are clustered in a distinct group away from neuronal changes.

Point 6. *Given the importance of TDP-43 function in ALS/FTD, I do not think it's sufficient to say that splicing analysis and analysis of cryptic exons are outside the scope of this study. There should be an effort to characterize alternative splicing events and particularly CEs known to be regulated by TDP-43. A recent paper (Gittings et al, PMID: 37466726) found that cryptic exons could be detected in snRNA-seq data from C9-ALS/FTD frontal cortex. Admittedly, this was most apparent in one sample that was deeply sequenced. However, assessing TDP-43 regulated splicing events in these data would provide another layer of information to compare with differential gene expression or chromatin accessibility between groups.*

Reply: We fully agree with the Reviewer and have added a significant amount of new experimental data to the study, which helped to increase the quality of the FANS-Seq dataset. Please see point 13 von Reviewer #2. We were now able to perform further analyses and demonstrate the inclusion of cryptic exons in TDP-43^{Low} cells. These data are provided in the new Fig 7-8 and Results sections 'Nuclear TDP-43 pathology selectively affects excitatory' and 'Transcriptional effects of TDP-43 pathology'.

Point 7. *In Figures 3 and 4, the authors mention that they identified transcriptional signatures common to previous analyses and known to be associated with ALS, but more detail would be helpful since the Figures do not highlight specific DEGs. Highly relevant findings (STMN2, etc.) would be useful to mention specifically.*

Reply: The Reviewer is right that specific well-known targets should be mentioned specifically to help the reader assess the agreement with known findings. We have explicitly shown *STMN2*, *NPTX2* and *EPHA4* in the revised manuscript (Results section ‘The ALS/ALS-FTD motor cortex is transcriptionally altered’, L 610-613).

Point 8. *I do not see a supplementary file that includes DEGs from the FANS-snrRNA-seq analysis. Please include this in any future submissions.*

Reply: We apologize for this inconsistency. We have included a list of the DEGs from the FANS-Seq analysis as the new Suppl. Tab. 12-21.

Point 9. *For the FANS-snrRNA-seq experiment, approximately 25% of nuclei sorted based on NeuN/RBFOX3 expression were characterized as ‘TDP-43-low.’ This far exceeds the expected % of cells exhibiting TDP-43 pathology in ALS-FTD motor cortex. This warrants explanation, as it is a major discrepancy and undermines the veracity of the entire FANS-snrRNA-seq data set.*

Reply: We apologize that we have not presented these data more clearly. Indeed, after sorting, we proceeded to single-nucleus RNA-sequencing on purpose with similar numbers of TDP-43 High and TDP-43 Low nuclei to enable meaningful statistical analyses (which have higher power if the two groups to compare are of similar size). In reality, many more TDP-43 High nuclei were found than TDP-43 Low, as expected. We added the quantification of 8 rounds of FANS-seq (6x from the final data and 2x from establishing runs) as the new Fig. 6 f and g. The numbers of TDP-43^{Low} nuclei after sorting were as expected – between 1

and 3 % of all neuronal nuclei. We thank the Reviewer for this remark which clearly improved the way the data is presented.

Point 10. *The TDP-43 low population also has a different proportion of cell types, with over-representation of LINC00507 excitatory neurons. Using snRNA-seq data from healthy controls, do these neurons (or a subset of these neurons) normally have lower TDP-43 expression? An additional control is necessary to ensure that the DEGs associated with TDP-43 depletion are not driven by differences in cell type proportion between the TDP-43 high and low groups. Could this be modeled by doing DEG analysis on two sets of randomly sampled nuclei comprised of the same cell type proportions as the TDP-43 high/low groups?*

Reply: We acknowledge the Reviewer's remarks, which are very important to ensure the quality of the data and support the conclusions. We have addressed all of them:

1) LINC00507 (now "L2-L3 Intratelencephalic") neurons do not have much lower TDP-43 expression than the other neuronal types, it is even higher than that of other excitatory neurons. We now show this in the new Fig. 6 b-d.

2) The fact that only 1-3 % of sorted nuclei are in the "TDP-43^{Low}" gate demonstrates that this is not a passive selection of excitatory neurons, which indeed have lower TDP-43 expression, but constitute 75% of all neuronal nuclei, not < 3%. In addition, *RBFOX3* (NeuN) expression is markedly lower in most types of inhibitory neurons than in excitatory neurons (new Fig. 6 b), but the NeuN signal of the TDP-43 High and TDP-43 Low gates runs "in parallel", demonstrating that the TDP-43^{Low} gate collects specifically nuclei with *disproportionately* low TDP-43 signal, not just with decreased expression.

3) The Reviewer is absolutely right and this is a central part of the revised study: the new data has enabled us to perform cell-type specific analyses to overcome the bias from different cell type proportions (new Fig. FANS 7-8, new Suppl. Tab. 17-21). Indeed, we show

that the previous studies and the cell-type unaware analysis are enriched in subtype markers, exactly as this Review expected (new Suppl. Fig. 19).

Minor points:

Point 11. *Overall, I noticed many small typos while reading the paper. I would recommend carefully re-reading the text for small typos. Examples:*

- *Fig S4 and S5,6: 'it parent cell population,' 'better suitable'*
- *Page 17: 'randomly samples genes'*
- *Figure 3: 'annotation of the sport'*

Reply: We apologize for these and have carefully revised the text to eliminate remaining typos.

Point 12. *The supplementary tables are not clearly labeled. If the file names are set by journal guidelines, the authors should include a brief description of the data within each file. For example, Suppl Tables 6-8 contain DEGs from pseudobulk analyses, but it is unclear which groups are being compared in each Table. All tables should include a description of the data within the file if the file names cannot be fully descriptive.*

Reply: The Reviewer is right, it is hard to tell when downloading and opening a supplementary data file which data is shown. We have included the description of the data to each single file. We thank the Reviewer for this suggestion!

Point 13. *Methods: FANS was performed "as previously described" and a link is provided, but I do not see a citation.*

Reply: We have added the appropriate citation to the text (Liu et al., Policichio et al.).

Point 14. *Results: patients were “diagnosed based on standardized criteria (methods)” but I do not see this information in the Methods section. This is a minor point because these samples do have confirmed TDP-43 pathology in the motor cortex.*

Reply: We have added this information also in the Methods section now (Methods section ‘Human samples and ethics’, L 115-117).

Point 15. Demographics information is limited to age and sex. Is information on patient race/ethnicity available?

Reply: Unfortunately, since the cohort is assembled from multiple centers, this data is not available in sufficient quality for the dataset. We have specified this in Methods section (Methods section ‘Human samples and ethics’, L 120-121).

Point 16. *The organization of Figure 2 should be improved. 2b and 2c blend into one another. Additionally, the Figure 2d legend refers to the data as %, but the figure represents the data as proportions (i.e., 0.25 versus 25%).*

Reply: We thank the Reviewer for this constructive suggestion. We have revised Figure 2 to enhance its consistency and readability.

Point 17. *Fig S5, S6, and S7 all have the same title. The authors should add details to define each figure (RNA-seq, ATAC-seq, etc.).*

Reply: This is correct, the titles are misleading. We have modified them to clarify what the separate figures show.

Point 18. *Page 24: “oligodendrocytes are affected by TDP-43 pathology in ALS-FTD, but not ALS” – this needs a citation.*

Reply: After further research, we realized that this statement was erroneous and have removed it from the text. We are very grateful to the Reviewer for pointing out this error.

Point 19. *Figure S10 G, H – the scale bars are set to highlight a lack of differences between C9 and sALS... but it seems that there are a few DEGs/DARs. It would be easier to interpret these panels if the axes were set lower (e.g., +/- 500).*

Reply: This is true, the aim of this visualization is to demonstrate the lack of differences b/w C9 and sALS-FTD. However, since it seems that the few DEGs/DARs may be interesting to the reader, we have added the numbers of DEGs above the bars in this chart.

Point 20. *Figure S11 uses “HC” as healthy control instead of Ctrl like other figures.*

Reply: We have corrected this inconsistency.

Point 21. *Figure S11 “Increasing dot size represents higher (red) or lower (blue) normalized enrichment score (‘NES’) for the respective term” - It is unclear to me if this is the same as saying blue dots represent downregulated genes, while red dots represent upregulated genes. If not, it would be more intuitive to see what pathways are enriched for up- vs down-regulated genes.*

Reply: We thank the Reviewer for this comment, which helped us to make the data presentation more clear. Indeed, red dots specify pathways which are enriched in the DEG list, and blue dots pathways which are decreased. We have specified this clearly in the revised Suppl. Fig. 14 legend.

Reviewer #4 (Remarks to the Author):

The authors used multi-modal single-nucleus profiling in the primary motor cortex of ALS and ALS-FTD to understand gene signatures and epigenetic regulatory mechanisms. The novelty of the study lies in their discovery of a distinct subset of transcriptomic changes not linked to TDP-43 pathology. Extensive bioinformatic data analyses were performed, providing a valuable single-nucleus dataset for the ALS field. However, the lack of a detailed molecular mechanism evaluation is a weak point of this study. The study could be significantly improved by performing experimental validation assays to support their hypothesis. Currently, no detail evaluation has been conducted for their findings.

Response to Reviewer #4:

Point 1. *The authors used multi-modal single-nucleus profiling in the primary motor cortex of ALS and ALS-FTD to understand gene signatures and epigenetic regulatory mechanisms. The novelty of the study lies in their discovery of a distinct subset of transcriptomic changes not linked to TDP-43 pathology. Extensive bioinformatic*

data analyses were performed, providing a valuable single-nucleus dataset for the ALS field. However, the lack of a detailed molecular mechanism evaluation is a weak point of this study. The study could be significantly improved by performing experimental validation assays to support their hypothesis. Currently, no detail evaluation has been conducted for their findings.

Reply: We thank the Reviewer for the effort. The Reviewer directly refutes the study on the sole grounds of lack of molecular validation studies. While we respect this view, we consider multi-omic approaches as well-established in the genomics field and they do provide novel mechanistic insights, which do not require the immediate translation to molecular biology assays in the same manuscript (e.g.

Nat Commun. 2024 Jul 10;15(1):5815 ⁴; Nat Commun. 2024 Dec 5;15(1):10609 ⁵; Nat Commun. 2025 Jan 2;16(1):319 ⁶. If more specifics are provided, we will be happy to provide wet lab validation wherever possible.

References

1. Li, J., *et al.* Divergent single cell transcriptome and epigenome alterations in ALS and FTD patients with C9orf72 mutation. *Nature communications* **14**, 5714 (2023).
2. Fajardo, C., *et al.* Von Economo neurons are present in the dorsolateral (dysgranular) prefrontal cortex of humans. *Neuroscience Letters* **435**, 215-218 (2008).
3. Brettschneider, J., *et al.* Stages of pTDP-43 pathology in amyotrophic lateral sclerosis. *Annals of neurology* **74**, 20-38 (2013).
4. Wang, Q., *et al.* Single cell transcriptomes and multiscale networks from persons with and without Alzheimer's disease. *Nature communications* **15**, 5815 (2024).

5. Rosario, S.R., *et al.* Integrative multi-omics analysis uncovers tumor-immune-gut axis influencing immunotherapy outcomes in ovarian cancer. *Nature communications* **15**, 10609 (2024).
6. Watson, B.R., *et al.* Spatial transcriptomics of healthy and fibrotic human liver at single-cell resolution. *Nature communications* **16**, 319 (2025).

We are grateful to the Reviewers for their effort and positive feedback after the manuscript revisions. Below is the point-to-point response to each new issue raised. We thank the Reviewers again.

Reviewer #1

(Remarks to the Author):

I am pleased to see that the authors have impressively strengthened their manuscript with the addition of the cryptic exon analysis and the deeper dive into the FANS-seq data. I have no further criticisms, though I implore the authors to make sure that all sequencing data, metadata, and count matrices are made publicly available at the point of publication.

(Remarks on code availability):

The code is exceptionally well organized. I congratulate the authors for being so transparent.

Response to Reviewer #1:

We thank the Reviewer for the positive feedback. The raw sequencing data, the processed sequencing data and the metadata have been made public on EGA and Zenodo.

Reviewer #2

(Remarks to the Author):

I greatly appreciate the author's willingness to address the many comments I made in the first round of reviews. This revised manuscript is greatly improved and I have a lot more confidence in the findings. Significant improvements include comparison of their results with the prior studies, and improved presentation and validation of their FANS dataset. I only have a few minor comments.

1. The "enrichment" of microglia in superficial layer 1 is interesting, but I would consider that this might not be an increase in superficial microglia but rather a decrease in other cellular elements. Given the paucicellular nature of layer 1, it is not surprising that other cell types are reduced which overall would appear like an increase of microglia. However, histologically, there is no accumulation of microglia in layer 1.

Reply: We thank the Reviewer for this useful insight. We have included it in the manuscript with a revision in the 'Results' section:

"Surprisingly, we observed an apparent enrichment of microglia in L1 in this highly validated dataset, but we could not identify whether this is an absolute enrichment of microglia or a relative enrichment due to lower numbers of other cell types without further studies."

2. I appreciate the wider maps shown in supplementary figure 13 for the rare Ecx FEZF NTNG1 cells. However, in some panels, there are dots that seem to indicate these are also found in superficial layers. This is likely due to the fact that different color scales are used for each pane (ranging from 0 to 6 versus 0 to 3). I suggest making all the panels use the same color scale, even if that means some panels do not identify these rare cells (as expected).

Reply: We thank the Reviewer for this constructive suggestion. Unfortunately, the function in the spatial imaging package used to plot these images does not include an option to specify the scale, so we were unable to plot all panels on the same scale.

3. The legend for supplemental figure #1 needs to be updated to include a description of panels e to k.

Reply: The Reviewer is right and we apologize for this error. We have updated the legend of Suppl. Fig. 1.

4. I appreciate that the FANS dot plots are now shown. For the gating, it appears that DAPI

singlets and doublets were included. If this is true, I would adjust the description in the methods where it says that only "single nuclei" were obtained.

Reply: The Reviewer highlights an important point. The singlet selection was based on the forward and sideward scatter. Although discrete "multiplets" can be seen in the DAPI plots, we restrained from filtering out the putative "doublets" in the DAPI gate, since brain cortical nuclei are very heterogenous in the DAPI signal and in our experience may appear as a bimodal population in DAPI: the small, compact nuclei of the glial cells and the much larger nuclei of the neurons. We have indicated this in the legend of Suppl. Fig. 20:

"Then, stained and intact nuclei from smaller and larger cells were selected based on DAPI signal..."

5. There is a typo on page 29 where "find" should be "found"

Reply: We thank the Reviewer for pointing this out. We have corrected the typing error:

"In addition, we observed that the counts of cryptic exon reads were mostly found in these 5 cell types, further..."

Reviewer #3

(Remarks to the Author):

The authors have provided substantive updates and clarifications to the original manuscript, which will make it more useful to the ALS research community.

Response to Reviewer #3:

We thank the Reviewer for the positive feedback and the constructive suggestions.